EMBO
Molecular Medicine

# Therapeutic targeting of *circ-CUX1*/EWSR1/MAZ axis inhibits glycolysis and neuroblastoma progression

Huanhuan Li[1,†], Feng Yang[1,†], Anpei Hu[1,†], Xiaojing Wang[2,†], Erhu Fang[1], Yajun Chen[3], Dan Li[1], Huajie Song[1], Jianqun Wang[1], Yanhua Guo[1], Yang Liu[1], Hongjun Li[3], Kai Huang[2], Liduan Zheng[2,3,*] (iD) & Qiangsong Tong[1,2,**] (iD)

## Abstract

Aerobic glycolysis is a hallmark of metabolic reprogramming in tumor progression. However, the mechanisms regulating glycolytic gene expression remain elusive in neuroblastoma (NB), the most common extracranial malignancy in childhood. Herein, we identify that CUT-like homeobox 1 (*CUX1*) and *CUX1*-generated circular RNA (*circ-CUX1*) contribute to aerobic glycolysis and NB progression. Mechanistically, p110 CUX1, a transcription factor generated by proteolytic processing of p200 CUX1, promotes the expression of enolase 1, glucose-6-phosphate isomerase, and phosphoglycerate kinase 1, while *circ-CUX1* binds to EWS RNA-binding protein 1 (EWSR1) to facilitate its interaction with MYC-associated zinc finger protein (MAZ), resulting in transactivation of MAZ and transcriptional alteration of *CUX1* and other genes associated with tumor progression. Administration of an inhibitory peptide blocking *circ-CUX1*-EWSR1 interaction or lentivirus mediating *circ-CUX1* knockdown suppresses aerobic glycolysis, growth, and aggressiveness of NB cells. In clinical NB cases, *CUX1* is an independent prognostic factor for unfavorable outcome, and patients with high *circ-CUX1* expression have lower survival probability. These results indicate *circ-CUX1*/EWSR1/MAZ axis as a therapeutic target for aerobic glycolysis and NB progression.

**Keywords** circular RNA; CUT-like homeobox 1; EWS RNA-binding protein 1; MYC-associated zinc finger protein; tumor progression
**Subject Categories** Cancer; RNA Biology

## Introduction

Neuroblastoma (NB), a malignant tumor arising from primitive neural crest, accounts for 15% of cancer-related mortality in childhood (Brodeur, 2003). For high-risk NB, the clinical outcome remains poor in despite of multimodal therapeutic approaches (Brodeur, 2003). To support their tumorigenecity and aggressiveness, tumor cells uptake and convert a large amount of glucose into lactic acid even in the presence of adequate oxygen, which is known as aerobic glycolysis or Warburg effect (Hanahan & Weinberg, 2011). Activation of oncogenes (*c-Myc*) or inactivation of tumor suppressors (*p53*) contributes to aberrant expression of transporters and metabolic enzymes of aerobic glycolysis (Shim *et al*, 1997; Bensaad *et al*, 2006; Yang *et al*, 2014). However, identification of transcriptional regulators for aerobic glycolysis in NB still remains to be determined.

Circular RNAs (circRNAs) are a novel class of endogenous noncoding RNAs that are generated from exons or introns, and may function as microRNA (miRNA) sponges, regulators of transcription and splicing, or partners of RNA-binding protein (RBP) (Lasda & Parker, 2014; Li *et al*, 2015b). For example, circRNA antisense to cerebellar-degeneration-related protein 1 (*Cdr1as*) harbors 70 binding sites for miR-7 to regulate its transport in neurons (Piwecka *et al*, 2017). A special class of exon–intron circRNAs, such as *circEIF3J* and *circPAIP2*, is predominantly localized in the nucleus and enhance transcription of their parental genes (Li *et al*, 2015b). In addition, intronic circRNAs, such as *ci-ankrd52*, are able to regulate transcription efficiency of parental genes by binding to RNA polymerase II (Zhang *et al*, 2013). However, the roles of circRNAs in aerobic glycolysis during tumor progression remain largely elusive.

In this study, we identify CUT-like homeobox 1 (CUX1) as a transcription factor facilitating aerobic glycolysis and tumor progression in NB. We also reveal the oncogenic functions of a *CUX1*-generated intron-containing circular RNA (*circ-CUX1*) in tumorigenesis and aggressiveness. Elevated *circ-CUX1* promotes the aerobic glycolysis, growth, and aggressiveness of NB cells by binding to EWS RNA-binding protein 1 (EWSR1) and facilitating its interaction with MYC-associated zinc finger protein (MAZ), resulting in MAZ transactivation and transcriptional alteration of *CUX1* and other genes associated with tumor progression, suggesting

1 Department of Pediatric Surgery, Union Hospital, Tongji Medical College, Huazhong University of Science and Technology, Wuhan, Hubei Province, China
2 Clinical Center of Human Genomic Research, Union Hospital, Tongji Medical College, Huazhong University of Science and Technology, Wuhan, Hubei Province, China
3 Department of Pathology, Union Hospital, Tongji Medical College, Huazhong University of Science and Technology, Wuhan, Hubei Province, China
*Corresponding author. Tel: +86 27 85726129; Fax: +86 27 85726821; E-mail: ld_zheng@hotmail.com
**Corresponding author. Tel: +86 27 85350762; Fax: +86 27 85726821; E-mail: qs_tong@hotmail.com
†These authors contributed equally to this work

*circ-CUX1*/EWSR1/MAZ axis as a therapeutic target for aerobic glycolysis and NB progression.

## Results

### CUX1 facilitates aerobic glycolysis and tumor progression

Comprehensive analysis of a microarray dataset (GSE16476) (Molenaar *et al*, 2012) of 88 NB cases identified 8 differentially expressed glycolytic genes (fold change > 2.0, $P < 0.05$) that were consistently associated with death, advanced international neuroblastoma staging system (INSS) stages, and clinical progression (Fig 1A). Similarly, we also found 52 transcription factors consistently associated with these clinical features (Fig 1A), which were subjective to further overlapping analysis with potential transcription factors regulating all of 8 glycolytic genes revealed by Genomatix program (http://www. genomatix.de/solutions/genomatix-software-suite.html). The results indicated CUX1 as the top transcription factor ranking by number of potential targets (Fig 1A). Higher transcript levels of *CUX1* isoform *p200* were noted in NB cell lines SH-SY5Y, IMR32, and SK-N-AS, while *p75* (Goulet *et al*, 2002) was expressed at very low levels (Appendix Fig S1A). Consistently, elevated levels of p200 CUX1 and its proteolytically processed isoform p110 were noted in these NB cells, cervical cancer HeLa cells, colon cancer LoVo cells, and prostate cancer PC-3 cells, than those of non-transformed normal MCF 10A cells (Appendix Fig S1A). However, both transcript and protein levels of CDP/cut alternatively spliced cDNA (*CASP*) (Gillingham *et al*, 2002) were not differently expressed between normal and tumor cells (Appendix Fig S1A). In an independent cohort of 54 primary NB tissues, the transcript levels of *p200 CUX1*, but not of *CASP*, were higher than those in normal fetal adrenal medulla ($P < 0.05$, Appendix Fig S1A), especially in cases with poor stroma ($P = 0.0002$) or advanced INSS stages ($P = 0.007$), without association with *MYCN* amplification ($P = 0.56$, Appendix Fig S1B).

Notably, ectopic expression or knockdown of *p200 CUX1* (referred as *CUX1*) increased and decreased the levels of p110 CUX1, enolase 1 (*ENO1*), glucose-6-phosphate isomerase (*GPI*), or phosphoglycerate kinase 1 (*PGK1*), but not of aldolase, fructose-bisphosphate A (*ALDOA*), glyceraldehyde-3-phosphate dehydrogenase (*GAPDH*), hexokinase 2 (*HK2*), lactate dehydrogenase A (*LDHA*), or pyruvate kinase M (*PKM*), in IMR32 and SH-SY5Y cells (Fig 1B and C, Appendix Fig S1C and D). Treatment with E64D, an inhibitor of

cathepsin L (Goulet *et al*, 2004), abolished the up-regulation of p110 CUX1, ENO1, GPI, and PGK1 induced by *CUX1* over-expression (Appendix Fig S1E). Ectopic expression of p110 CUX1 increased the levels of *ENO1*, *GPI*, or *PGK1* in IMR32 cells (Appendix Fig S1E). Meanwhile, knockdown of *CASP*, a Golgi-localized *CUX1* isoform (Gillingham *et al*, 2002), did not affect the transcript and protein levels of these glycolytic genes in SH-SY5Y cells (Appendix Fig S1F and G). The CUX1 enrichment and promoter activity of *ENO1*, *GPI*, and *PGK1* were increased and decreased by p110 CUX1 over-expression, *CUX1* knockdown, or E64D treatment in IMR32 and SH-SY5Y cells, respectively (Appendix Fig S1H, Fig 1D and E). Over-expression of p110 CUX1 increased the extracellular acidification rate (ECAR) and reduced the oxygen consumption rate (OCR) in IMR32 cells, while *CUX1* knockdown or E64D treatment significantly attenuated the glycolytic process (Fig 1F and G). Accordingly, p110 CUX1 over-expression, *CUX1* knockdown, or E64D treatment increased and decreased the glucose uptake, lactate production, ATP levels, anchorage-independent growth, and invasion of IMR32 cells, respectively (Appendix Fig S2A–D). Treatment with 2-deoxy-glucose (2-DG), an established glycolysis inhibitor (Zhang *et al*, 2014), abolished the increase in these features of IMR32 cells induced by p110 CUX1 over-expression (Appendix Fig S2A–D). In public datasets, there was positive expression correlation between *CUX1* and *ENO1*, *GPI*, or *PGK1* in NB, colon cancer, or prostate cancer tissues (Appendix Fig S2E), and their levels were associated with poor survival of tumor patients (Appendix Fig S3). Multivariate Cox regression analysis revealed *CUX1* as an independent prognostic factor [hazard ratio = 2.105, 95% confidence interval = 1.087–3.243, $P = 0.038$] for poor survival of NB patients. These findings indicated that CUX1 was a transcription factor facilitating aerobic glycolysis and tumor progression.

### Circ-CUX1 is up-regulated in NB tissues and cell lines

The copy number of *CUX1* gene, locating at chr7: 101460882-101901513, was neither significantly altered in NB (Appendix Fig S4A) nor associated with death, *MYCN* amplification, INSS stages, or survival of NB cases derived from Oncogenomics database (Appendix Fig S4A and B). There were no genetic variants of *CUX1* gene in 563 NB cases of public datasets (Appendix Fig S4C). Among 37 potential circRNAs generated from *CUX1* gene in circBase (Glazar *et al*, 2014), 15 had more than 2 read scores, while further RT–PCR validation revealed 7 detectable circRNAs in IMR32 cells (Fig EV1A

---

**Figure 1. CUX1 facilitates aerobic glycolysis and tumor progression in NB.**

A Venn diagram indicating the identification of differentially expressed glycolytic genes and transcription factors (fold change > 2.0, Student's *t*-test, $P < 0.05$) in 88 NB cases (GSE16476), and over-lapping analysis with potential transcription factors regulating glycolytic genes revealed by Genomatix program.

B, C Real-time qRT–PCR (B, normalized to β-actin, $n = 5$) and Western blot (C) assays revealing the expression of *CUX1* and glycolytic genes in IMR32 cells stably transfected with empty vector (mock), *p200 CUX1*, scramble shRNA (sh-Scb), sh-CUX1 #1, or sh-CUX1 #2. Student's *t*-test, one-way ANOVA, *$P < 0.05$ versus mock or sh-Scb.

D, E ChIP and qPCR using Flag and CUX1 antibodies (D) and dual-luciferase (E) assays indicating the p110 CUX1 enrichment and promoter activity of *ENO1*, *GPI*, and *PGK1* in IMR32 and SH-SY5Y cells stably transfected with mock, Flag-tagged *p110 CUX1*, sh-Scb, sh-CUX1 #1, or sh-CUX1 #2, and those treated with E64D (10 μM) for 24 h ($n = 5$). Student's *t*-test, one-way ANOVA, *$P < 0.05$ versus mock, sh-Scb, or DMSO.

F, G Seahorse tracing curves (F), ECAR and OCR (G) of IMR32 cells stably transfected with mock, *p110 CUX1*, sh-Scb, sh-CUX1 #1, or sh-CUX1 #2, and those treated with E64D (10 μM) for 24 h ($n = 5$). Student's *t*-test, one-way ANOVA, *$P < 0.05$ versus mock, sh-Scb, or DMSO.

Data information: Data are presented as mean ± SEM. Exact *P* values are specified in Appendix Table S4.
Source data are available online for this figure.

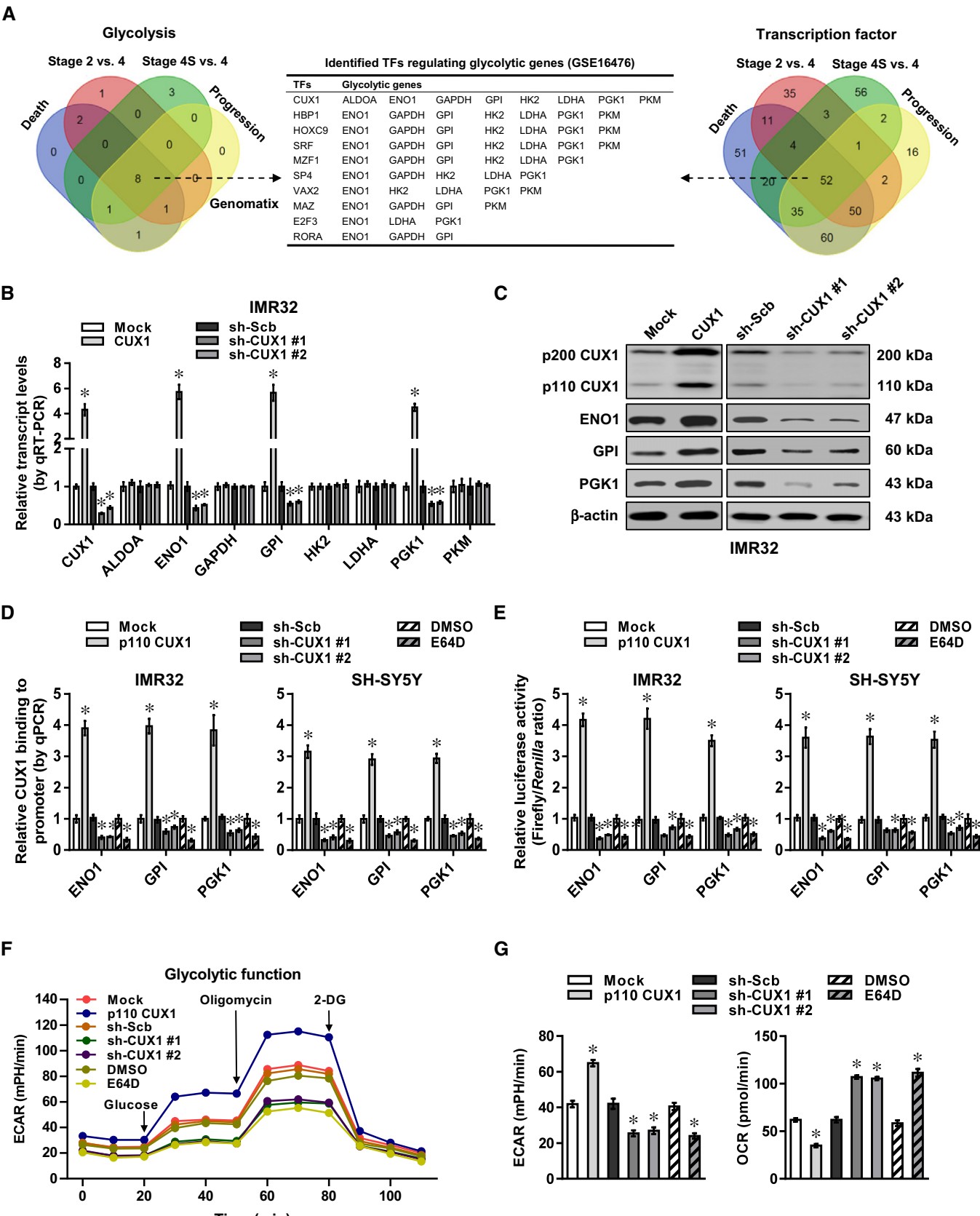

**Figure 1.**

and B). Three of them were differentially expressed between normal fetal adrenal medulla and NB tissues (Fig EV1C). Knockdown of *hsa_circ_0132813* (termed as *circ-CUX1*), but not of *hsa_circ_0132817* or *hsa_circ_0005253*, attenuated the CUX1 transactivation in IMR32 and SH-SY5Y cells (Fig EV1D). The 393-nt *circ-CUX1*, consisting of exon 2 and partial intron 2 of *CUX1* (Fig 2A), was detected by RT–PCR with divergent primers and Sanger sequencing (Fig 2B), and its expression levels were significantly elevated in many tumor cell lines (Fig 2C and D). Endogenous *circ-CUX1* was resistant to RNase R digestion (Fig 2D) and localized within nucleus of IMR32 cells, which was further confirmed by ectopic expression of *circ-CUX1* (Fig 2D–F). Notably, *circ-CUX1* levels were higher in tissues of NB, colon cancer, and prostate cancer, than normal fetal adrenal medulla or adjacent normal tissues (Fig 2G). In addition, *circ-CUX1* levels were positively correlated with those of *CUX1* in tissues of NB ($R = 0.590$, $P < 0.0001$), colon cancer ($R = 0.868$, $P < 0.0001$), or prostate cancer ($R = 0.619$, $P = 0.0023$; Fig 2G). In 54 primary NB tumors, higher *circ-CUX1* levels were observed in cases with poor stroma ($P = 0.0083$) or advanced INSS stages ($P = 0.0017$), without association with *MYCN* amplification ($P = 0.1532$; Fig EV1E). Patients with high *circ-CUX1* expression had lower survival probability (Fig EV1F). These results indicated that *circ-CUX1* was up-regulated in NB tissues and cell lines.

### Circ-CUX1 enhances CUX1 expression at transcriptional level

To investigate the effects of *circ-CUX1* on expression of parental gene *CUX1*, *circ-CUX1* or two independent short hairpin RNAs (shRNAs) targeting junction site of *circ-CUX1* (sh-circ-CUX1) were stably transfected into tumor cell lines. Transfection of *circ-CUX1* vector, but not of that with mutant back-splicing elements (*circ-CUX1*-Mut), resulted in obvious *circ-CUX1* production in IMR32 cells, which was resistant to RNase R digestion (Appendix Fig S5A). Transfection of sh-circ-CUX1 #1 and sh-circ-CUX1 #2 increased the enrichment of Argonaute 2 (AGO2) on *circ-CUX1*, but not on *CUX1* mRNA, in IMR32, SH-SY5Y, LoVo, and PC-3 cells (Appendix Fig S5B). In a luciferase reporter-based assay monitoring shRNA specificity (Bramsen *et al*, 2010), transfection of sh-circ-CUX1 #1 or sh-circ-CUX1 #2 decreased the activity of *circ-CUX1* reporter, without impact on that of *CUX1* reporter (Appendix Fig S5C). Notably, stable transfection of *circ-CUX1*, but not of *circ-CUX1*-Mut, into IMR32, SH-SY5Y, LoVo, and PC-3 cells resulted in its over-expression (Fig EV2A), increased *CUX1* promoter activity (Fig EV2B), and elevated levels of *CUX1* isoforms *p200* and *p110*, but not of *CASP*

(Figs EV2C and 2H). The stability of *CUX1* mRNA was not altered in IMR32 cells stably transfected with *circ-CUX1* (Fig EV2D). Meanwhile, stable knockdown of *circ-CUX1* led to decrease in *CUX1* promoter activity and expression of *p200* and *p110* in tumor cells, without effects on *CUX1* mRNA stability or *CASP* levels (Figs EV2A–D and 2H). These results illustrated that *circ-CUX1* enhanced *CUX1* expression at transcriptional level in tumor cells.

### Circ-CUX1 exerts an oncogenic role in tumor progression

We further observed the potential effects of *circ-CUX1* on biological features of tumor cells. The ECAR was increased and decreased in IMR32, SH-SY5Y, LoVo, and PC-3 cells stably transfected with *circ-CUX1* or sh-circ-CUX1, along with reduced and enhanced OCR, while transfection of *circ-CUX1*-Mut did not affect these features (Fig EV2E). Notably, ectopic expression of *circ-CUX1* increased the glucose uptake, lactate production, and ATP levels of IMR32 cells, which was attenuated by 2-DG treatment (Appendix Fig S6A). Stable over-expression or knockdown of *circ-CUX1* increased and decreased the anchorage-independent growth and invasion of IMR32 and SH-SY5Y cells, respectively (Fig 3A and B). Consistently, stable transfection of *circ-CUX1* or sh-circ-CUX1 #1 into IMR32 cells resulted in a significant increase or decrease in growth, tumor weight, Ki-67 proliferation index, CD31-positive microvessels, glucose uptake, lactate production, and ATP levels of subcutaneous xenograft tumors in nude mice (Fig 3C–E). Athymic nude mice treated with tail vein injection of IMR32 cells with stable over-expression or knockdown of *circ-CUX1* displayed more or less lung metastatic colonies, with lower or greater survival probability, respectively (Fig 3F). These results indicated that *circ-CUX1* exerted an oncogenic role in tumorigenesis and aggressiveness.

### Circ-CUX1 directly interacts with EWSR1 protein in NB cells

To explore the protein partner of *circ-CUX1*, RNA pull-down was performed using biotin-labeled probes generated by ligation of linear transcript *in vitro* (Petkovic & Muller, 2015) or synthesized as oligonucleotides targeting junction site (Fig 4A). Mass spectrometry revealed 47 proteins consistently pulled down by exogenous *circ-CUX1* and antisense probe targeting endogenous *circ-CUX1*, but not by linear transcript or sense probe, and 18 of them were RBPs defined by RBPDB (http://rbpdb.ccbr.utoronto.ca). Further comprehensive analysis of protein interacting with transcription factors of *CUX1* promoter revealed by Genomatix and BioGRID database

---

**Figure 2.** **Circ-CUX1 is up-regulated and enhances *CUX1* expression in NB.**

A       Schematic illustration showing the generation of *circ-CUX1* from *CUX1*.
B       RT–PCR or PCR assay revealing the amplification of *circ-CUX1* from cDNA or genomic DNA (gDNA) of IMR32 and HeLa cells, with validation by Sanger sequencing.
C, D    Real-time qRT–PCR (C, normalized to β-actin, $n = 6$) and Northern blot (D) indicating the *circ-CUX1* levels in cell lines and IMR32 cells transfected with empty vector (mock) or *circ-CUX1* and treated with RNase R (3 U µg$^{-1}$). One-way ANOVA, *$P < 0.05$ versus HEK293.
E, F    Real-time qRT–PCR (E, normalized to β-actin) and RNA-FISH with antisense junction probe and RNase R (3 U µg$^{-1}$) treatment (F) showing the distribution and localization (arrowheads) of *circ-CUX1* in IMR32 cells stably transfected with mock or *circ-CUX1* ($n = 5$), using *GAPDH* and *U1* as controls. Scale bar: 10 µm.
G       Real-time qRT–PCR assay indicating *circ-CUX1* expression (normalized to β-actin) and its correlation with *CUX1* levels (Pearson's correlation coefficient) in tumor tissues, normal fetal adrenal medulla (FAM), or normal counterparts. Student's *t*-test, **$P < 0.01$ versus FAM or normal.
H       Western blot showing the CUX1 levels in tumor cells stably transfected as indicated.

Data information: Data are presented as mean ± SEM. Exact *P* values are specified in Appendix Table S4.

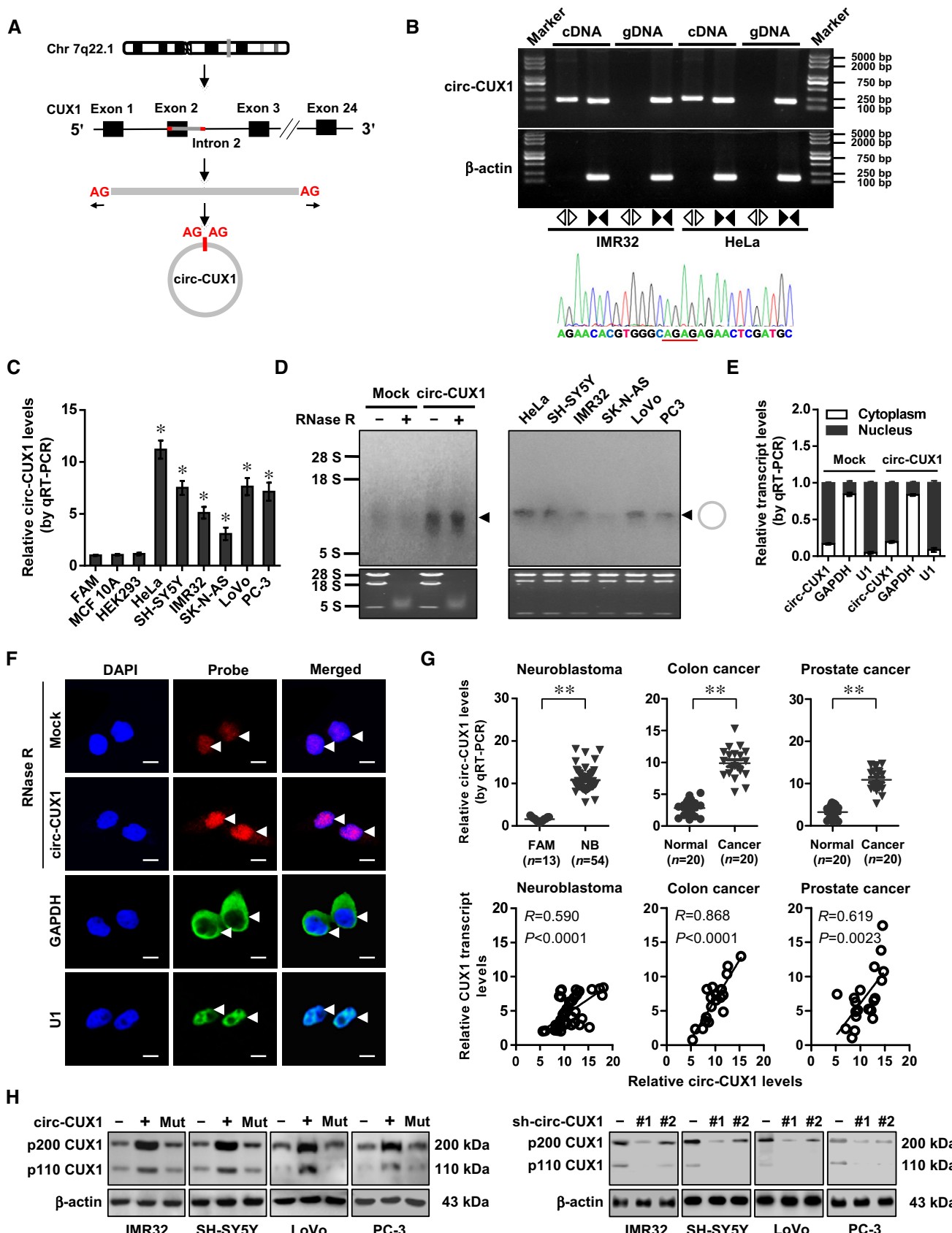

**Figure 2.**

(https://thebiogrid.org) indicated three potential *circ-CUX1*-inter-acting partners (Fig 4A), including EWSR1, ELAV-like RNA-binding protein 1 (ELAVL1), and synaptotagmin-binding cytoplasmic RNA-interacting protein (SYNCRIP). Further validating RNA pull-down assay using *circ-CUX1* probes revealed the specific enrichment of endogenous or exogenous *circ-CUX1*, but not *p200 CUX1* or *CASP* transcript, and the physical interaction of *circ-CUX1* with EWSR1, but not with ELAVL1 or SYNCRIP, in non-transfected IMR32 cells (Fig 4B). Endogenous binding of EWSR1 protein to *circ-CUX1*, but not to *p200 CUX1* or *CASP* transcript, was also observed in SH-SY5Y cells, which was facilitated by transfection of *circ-CUX1* (Fig 4C). In NB cell lines SH-SY5Y, SK-N-AS, BE(2)-C, and IMR32, PCR assay indicated no fusion of *EWSR1* with *Fli1* gene (Appendix Fig S6B). Three-dimensional (3D) confocal images of dual RNA fluorescence *in situ* hybridization (RNA-FISH) and immunofluorescence assay confirmed endogenous co-localization of *circ-CUX1* and EWSR1 in IMR32 cells, which was facilitated by transfection of *circ-CUX1* (Fig 4D and Movie EV1). Consistently, RNA electrophoretic mobility shift assay (EMSA) showed that *circ-CUX1* interacted with EWSR1 protein within nuclear extracts of SH-SY5Y cells (Fig 4E). The RNA recognition motif (RRM) domain [361–447 amino acids (aa)], but not amino- or carboxyl-terminus, of glutathione S-transferase (GST)-tagged EWSR1 protein was necessary for its interaction with *circ-CUX1*, but not with *p200 CUX1* or *CASP* transcript (Fig 4F). Mutation of 394–397 or 406–410 aa of RRM domain, potential interacting regions analyzed by catRAPID (Agostini *et al*, 2013), abolished the interaction of EWSR1 with *circ-CUX1* (Fig 4F). These results suggested that *circ-CUX1* directly interacted with EWSR1 protein in NB cells.

### Circ-CUX1 facilitates EWSR1-mediated MAZ transactivation

To further investigate target genes of *circ-CUX1*, RNA sequencing (RNA-seq) was performed and revealed 781 up-regulated and 434 down-regulated genes (fold change > 1.5, $P < 0.05$) in IMR32 cells upon *circ-CUX1* over-expression (Fig 5A). Transcription factors regulating these genes were analyzed by ChIP-X program (Lachmann *et al*, 2010), which revealed top five potential ones, including E12, lymphoid enhancer-binding factor 1 (LEF1), MAZ, nuclear factor of activated T cells (NFAT), and specificity protein 1 (SP1) (Fig 5B). Further over-lapping analysis with EWSR1-interacting proteins derived from BioGRID database revealed that MAZ protein was the only transcription factor involved in this process (Fig 5B).

Endogenous interaction between EWSR1 and MAZ was observed in IMR32 cells (Appendix Fig S6C). The RRM domain (361–448 aa), but not transactivation domain (TAD), Arg-Gly-Gly (RGG) 1, RGG2, or RGG3 domain, of Myc-tagged EWSR1 was necessary for its inter-action with MAZ protein (Appendix Fig S6D). Meanwhile, zinc fin-ger (ZNF) domain (198–477 aa), but not N-terminus or C-terminus, of Flag-tagged MAZ was necessary for its interaction with EWSR1 (Appendix Fig S6E). Ectopic expression or knockdown of *circ-CUX1* increased and decreased the interaction between EWSR1 and MAZ in IMR32 and SH-SY5Y cells, respectively (Fig 5C and D, and Appendix Fig S6F).

Notably, higher *MAZ* levels were observed in NB tissues than those in normal fetal adrenal medulla ($P < 0.0001$), especially in those with poor stroma ($P = 0.0205$) or advanced INSS stages ($P = 0.0097$), without association with *MYCN* amplification ($P = 0.6445$, Appendix Fig S7A). Among 60 *MAZ* target genes derived from RNA-seq results and ChIP-X analysis, the expression of *CUX1*, S100 calcium-binding protein A9 (*S100A9*), mucin 4 (*MUC4*), Kruppel-like factor 10 (*KLF10*), or thioredoxin-interacting protein (*TXNIP*) was significantly correlated with that of *MAZ* in 54 NB cases (Appendix Fig S7B). In addition, higher expression of *EWSR1*, *MAZ*, *S100A9*, or *MUC4* and lower expression of *KLF10* or *TXN1P* were associated with poor survival of NB patients (GSE16476, Appendix Fig S7C). In RNA pull-down and chromatin isolation by RNA purification (ChIRP) (Chu & Chang, 2016) assays using biotin-labeled *circ-CUX1* junction probe, *circ-CUX1* was associated with EWSR1 and MAZ protein, and promoters of target genes (*CUX1*, *S100A9*, *MUC4*, *KLF10*, or *TXNIP*), but not with transcripts of down-stream genes in SH-SY5Y cells (Appendix Fig S8A). Ectopic expres-sion or knockdown of *circ-CUX1* enhanced and reduced the binding of MAZ to these target gene promoters in IMR32 and SH-SY5Y cells, while silencing or over-expression of *EWSR1* abolished these effects (Fig 5E and Appendix Fig S8B). The activity of wild-type *CUX1* promoter, but not of that with mutant MAZ-binding site, was increased and decreased by ectopic expression or knockdown of *circ-CUX1* (Fig 5F and Appendix Fig S8C). In addition, the levels of *CUX1*, *S100A9*, *MUC4*, *KLF10*, or *TXNIP* were significantly altered in IMR32 and SH-SY5Y cells stably transfected with *circ-CUX1* or sh-circ-CUX1 #1 (Fig 5G and H, Appendix Fig S8D and E). Knockdown or ectopic expression of *EWSR1* or *MAZ* rescued tumor cells from these changes (Fig 5F–H, Appendix Figs S8C–E and S9A). These data indicated that *circ-CUX1* facilitated EWSR1-mediated MAZ transacti-vation and transcriptional alteration of target genes in NB cells.

---

**Figure 3. *Circ-CUX1* exerts an oncogenic role in tumorigenesis and aggressiveness.**

A, B   Soft agar (A) and Matrigel invasion (B) assays showing the anchorage-independent growth and invasion capability of IMR32 and SH-SY5Y cells stably transfected with empty vector (mock), *circ-CUX1*, scramble shRNA (sh-Scb), or sh-circ-CUX1 ($n = 5$). Scale bars: 100 μm. Student's *t*-test, one-way ANOVA, *$P < 0.05$ versus mock or sh-Scb.

C   Representative fluorescence images, *in vivo* growth curve, and weight at the end points of subcutaneous xenograft tumors formed by IMR32 cells stably transfected as indicated in nude mice ($n = 5$ for each group). Student's *t*-test, one-way ANOVA, *$P < 0.05$ versus mock or sh-Scb.

D, E   Immunohistochemical staining showing the expression of Ki-67 and CD31 (D) and glucose uptake, lactate production, and ATP levels (E) within subcutaneous xenograft tumors formed by IMR32 cells stably transfected as indicated ($n = 5$ for each group). Scale bars: 100 μm. Student's *t*-test, **$P < 0.01$ versus mock or sh-Scb.

F   Representative images, HE staining (arrowheads), quantification of lung metastatic colonization, and Kaplan–Meier curves of nude mice treated with tail vein injection of IMR32 cells stably transfected as indicated ($n = 5$ for each group). Scale bar: 100 μm. Student's *t*-test, **$P < 0.01$ versus mock or sh-Scb. Log-rank test for survival comparison.

Data information: Data are presented as mean ± SEM. Exact *P* values are specified in Appendix Table S4.

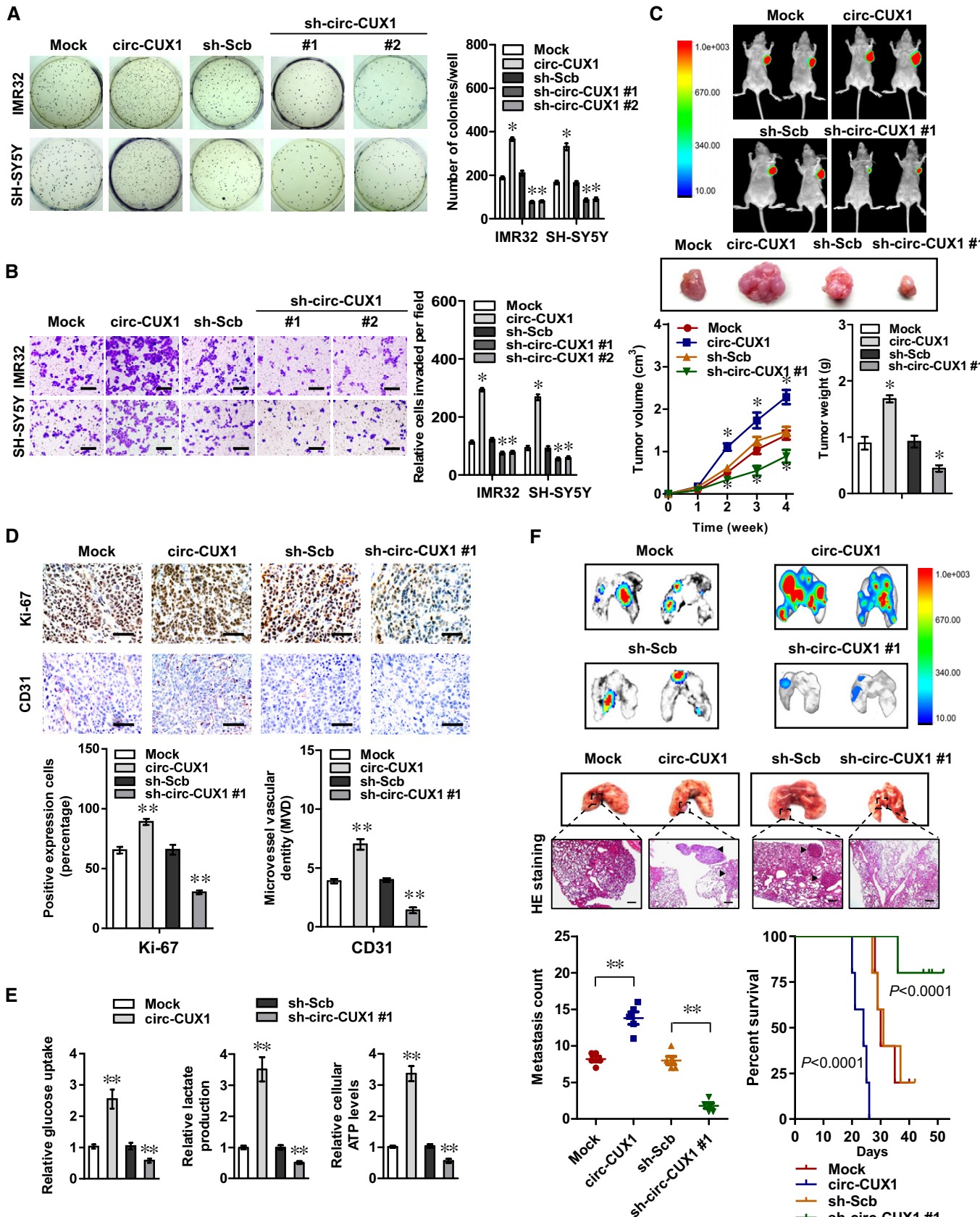

Figure 3.

### Therapeutic peptide blocking the *circ-CUX1*-EWSR1 interaction

Based on the importance of RRM domain (especially 394–397 or 406–410 aa) of EWSR1 in interacting with *circ-CUX1*, we further designed a cell-penetrating peptide, named as EWSR1 inhibitory peptide of 22 amino acids (EIP-22), that might potentially block *circ-CUX1*-EWSR1 interaction (Fig 6A). Treatment of SH-SY5Y cells with EIP-22 resulted in its obvious aggregation within the nucleus (Fig 6B). Biotin-labeled peptide pull-down assay revealed the binding of EIP-22 to endogenous *circ-CUX1* in SH-SY5Y cells (Appendix Fig S9B). In addition, EIP-22 treatment reduced the interaction between *circ-CUX1* and EWSR1, but not that of *pri-miR-222* and EWSR1 (Ouyang *et al*, 2017) or *circACC1* and AMP-activated protein kinase beta 1 (AMPKβ1) (Li *et al*, 2019; Fig 6C and D). Administration of EIP-22 inhibited the viability, anchorage-independent growth, and invasion of SH-SY5Y cells (Appendix Fig S9C, Fig 6E and F), with alteration of *circ-CUX1* downstream gene expression (Appendix Fig S9D). In contrast, EIP-22 treatment resulted in no significant alteration in the viability of MCF 10A, non-transformed normal cells with very low *circ-CUX1* expression (Fig 2C and Appendix Fig S9C). Notably, EIP-22 treatment synergized the suppressing effects of glycolysis inhibitors, 2-DG and 3-bromopyruvate (3-BP) (Cardaci *et al*, 2012; Zhang *et al*, 2014), on the viability, growth, and invasion of IMR32 and SH-SY5Y cells (Appendix Fig S9E–G). Intravenous administration of EIP-22 significantly reduced the growth, tumor weight, Ki-67 proliferation index, and CD31-positive microvessels, altered *circ-CUX1* target gene expression, and decreased the glucose uptake, lactate production, and ATP levels in subcutaneous xenograft tumors formed by injection of SH-SY5Y cells (Fig 6G and Appendix Fig S10A–C). Moreover, administration of EIP-22 via tail vein reduced the lung metastatic colonies and prolonged the survival time of athymic nude mice received tail vein injection of SH-SY5Y cells (Fig 6H). These data suggested that EIP-22 suppressed NB progression by blocking *circ-CUX1*-EWSR1 interaction.

### Therapeutic lentivirus-mediated *circ-CUX1* knockdown *in vivo*

We further explored the therapeutic efficiencies of *circ-CUX1* knockdown on athymic nude mice bearing xenograft tumors formed by subcutaneous or tail vein injection of IMR32 cells. Lentivirus-mediated knockdown of *circ-CUX1* significantly reduced the growth, tumor weight, Ki-67 proliferation index, and CD31-positive microvessels of subcutaneous xenograft tumors (Appendix Fig S11A and B), with altered expression of *circ-CUX1* target genes (Appendix Fig S11C). The glucose uptake, lactate production, and ATP levels were significantly decreased in xenograft tumors of nude mice received tail vein injection of lentivirus carrying sh-circ-CUX1 (Appendix Fig S11D). In addition, administration of lentivirus carrying sh-circ-CUX1 #1 decreased the lung metastatic colonies and prolonged the survival time of nude mice (Appendix Fig S11E). These results indicated that lentivirus-mediated *circ-CUX1* knockdown inhibited aerobic glycolysis and NB progression *in vivo*.

## Discussion

Recent studies show that although LDHA and LDHB promote tumorigenicity, they are dispensable for aerobic glycolysis in NB (Dorneburg *et al*, 2018), suggesting the involvement of other glycolytic genes in this process. So far, interrogative screening of transcriptional regulators of aerobic glycolysis in NB remains unknown. In this study, we identified CUX1 as a transcription factor facilitating the expression of glycolytic genes *ENO1*, *GPI*, and *PGK1* in NB. We demonstrate that *circ-CUX1* interacts with EWSR1 protein to increase MAZ transactivation, which subsequently regulates the transcription of *CUX1* and other genes associated with tumor progression *in cis* and *in trans* (Fig 6I), such as *S100A9* (Lim *et al*, 2016), *MUC4* (Rowson-Hodel *et al*, 2017), *KLF10* (Weng *et al*, 2017), and *TXNIP* (Shen *et al*, 2015). The discovery of *circ-CUX1*/EWSR1/MAZ axis represents a promising step for therapeutic intervention against tumors.

CUX1 is a transcription factor involved in embryonic development (Michl *et al*, 2005; Harada *et al*, 2008) and regulates cellular proliferation, migration, and epithelial-to-mesenchymal transition, suggesting its emerging roles in tumorigenesis and aggressiveness (Michl *et al*, 2005). Elevated *CUX1* expression has been documented in many tumors and is associated with poor survival of patients (Liu *et al*, 2013). Full-length p200 CUX1 binds rapidly but only transiently to DNA (Liu *et al*, 2013), while its proteolytic product p110 isoform activates gene transcription (Harada *et al*, 2008; Kedinger *et al*, 2009). In this study, our results indicated that *CUX1* was an independent prognostic marker for progression and poor outcome of NB. In addition, p110 CUX1 promoted the expression of target genes

---

**Figure 4.  *Circ-CUX1* directly interacts with EWSR1 protein in NB cells.**

A  Schematic illustration, Coomassie Blue staining, and Venn diagram showing the differential proteins pulled down by biotin-labeled linear or circular exogenous *circ-CUX1*, sense (S) or antisense (AS) probe targeting junction site of endogenous *circ-CUX1* from IMR32 cells, and over-lapping analysis with RBP and proteins interacting with potential transcription factors of *p200 CUX1* revealed by Genomatix program and BioGRID database.

B  Western blot (upper panel) and RT–PCR (lower panel) assays indicating the proteins and transcripts pulled down by biotin-labeled linear or circular exogenous *circ-CUX1*, sense (S) or antisense (AS) probe targeting junction site of endogenous *circ-CUX1* from IMR32 cell lysates, using AS probes of *p200 CUX1* or *CASP* as controls.

C  RIP and real-time qRT–PCR assays revealing the interaction of EWSR1 with *circ-CUX1*, *p200 CUX1*, or *CASP* in SH-SY5Y cells and those stably transfected with empty vector (mock) or *circ-CUX1* (n = 5). Student's *t*-test, *P < 0.05 versus IgG; $^{\Delta}$P < 0.01 versus mock.

D  3D confocal images of dual RNA-FISH and immunofluorescence staining assay showing the co-localization of *circ-CUX1* and EWSR1 in IMR32 cells stably transfected with mock or *circ-CUX1*. Scale bar: 10 μm.

E  RNA EMSA determining the interaction between biotin-labeled *circ-CUX1* probe and EWSR1 protein within nuclear extracts of SH-SY5Y cells (arrowheads).

F  *In vitro* binding assay depicting the recovered *circ-CUX1*, *p200 CUX1*, or *CASP* detected by RT–PCR (upper panel) after incubation with GST-tagged recombinant EWSR1 protein validated by Western blot (lower panel).

Data information: Data are presented as mean ± SEM. Exact *P* values are specified in Appendix Table S4.
Source data are available online for this figure.

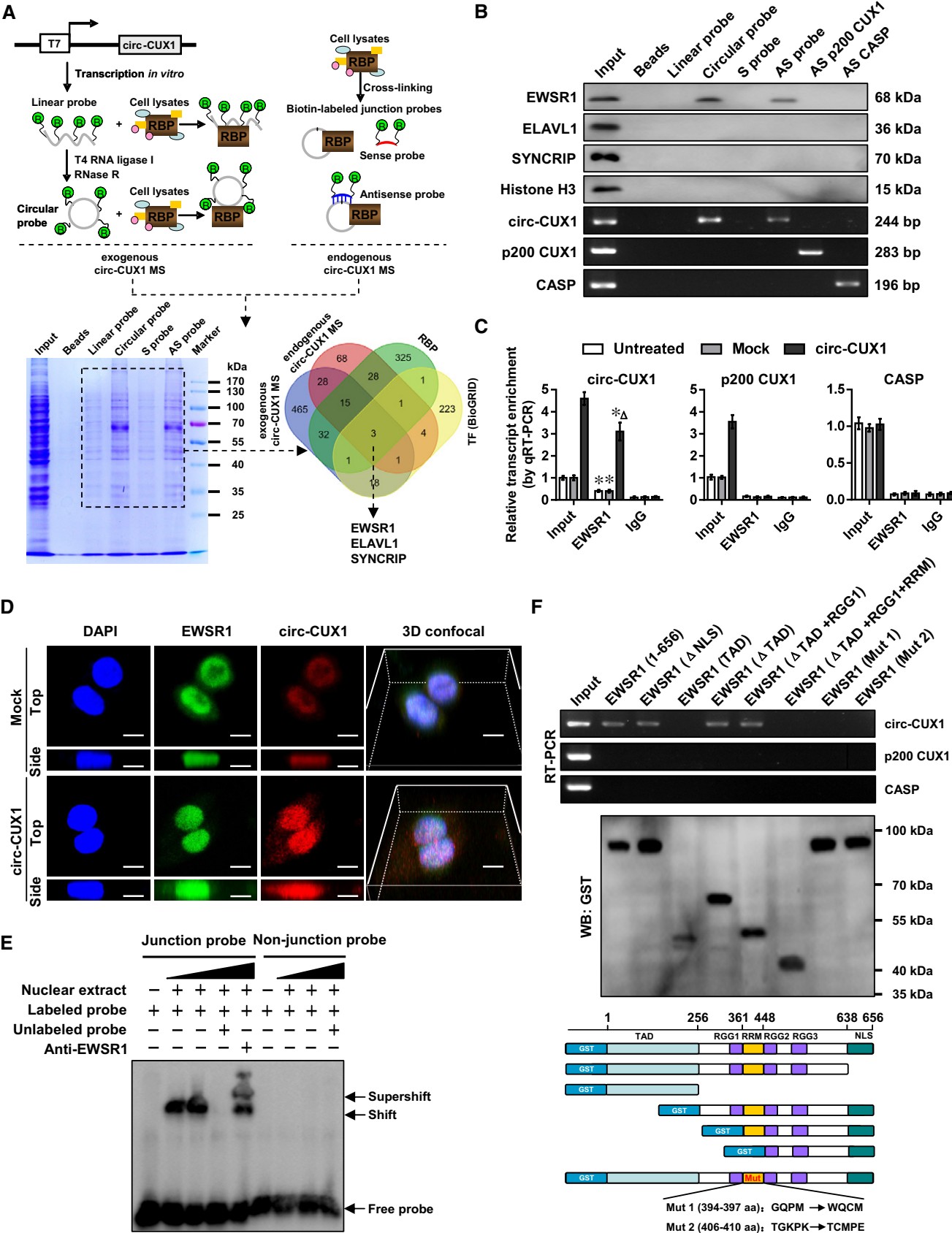

**Figure 4.**

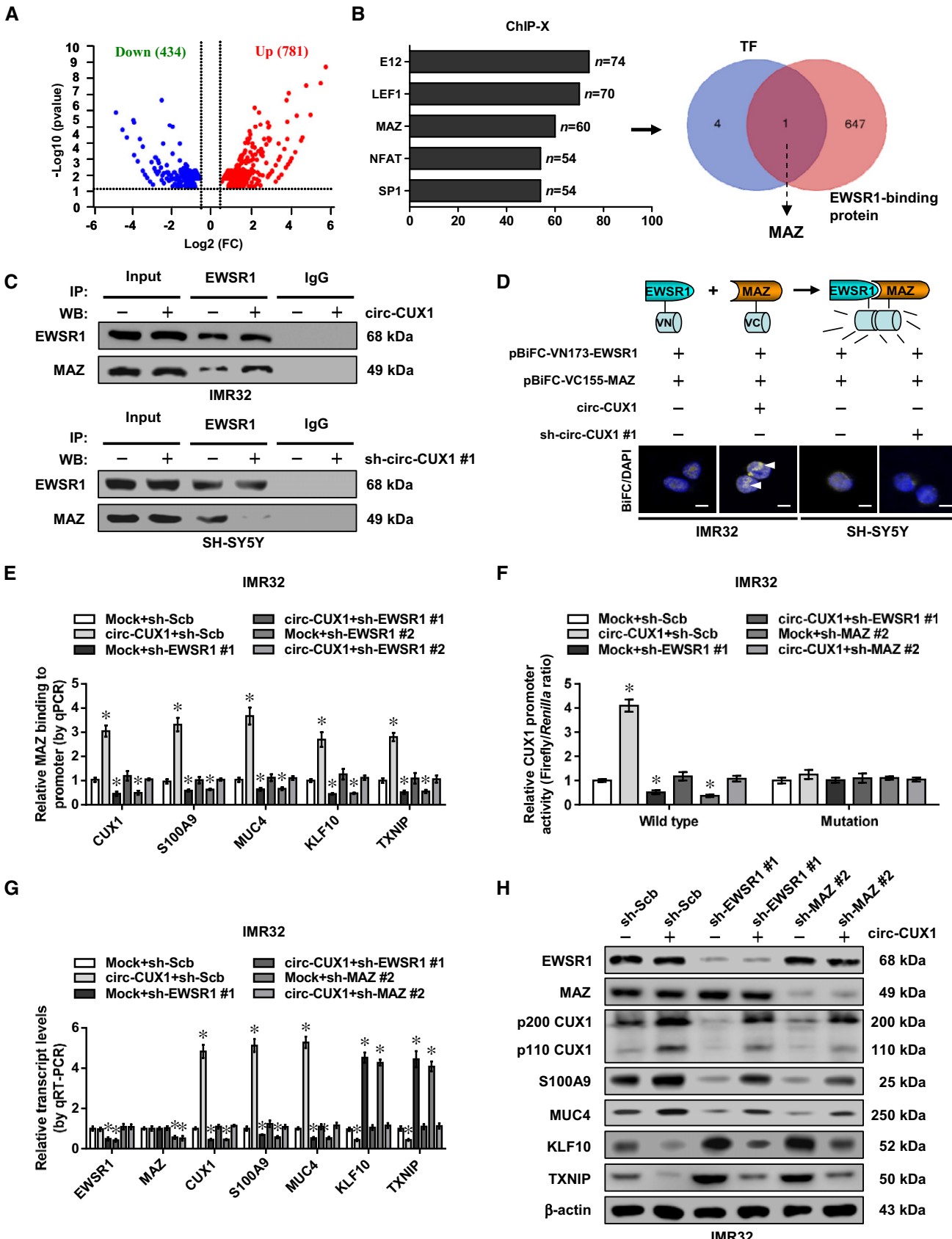

**Figure 5.**

**Figure 5.** ***Circ-CUX1*** **facilitates EWSR1-mediated MAZ transactivation in NB cells.**

A  Volcano plots indicating RNA-seq results of 781 up-regulated and 434 down-regulated genes in IMR32 cells upon stable *circ-CUX1* over-expression (fold change > 1.5, *P* < 0.05).

B  ChIP-X analysis (left panel) showing top five transcription factors regulating the altered genes, and Venn diagram (right panel) indicating the identification of MAZ by over-lapping analysis of five transcription factors and EWSR1-interacting proteins derived from BioGRID database.

C  Co-IP and Western blot assays showing the interaction between EWSR1 and MAZ in IMR32 and SH-SY5Y cells stably transfected with *circ-CUX1* or sh-circ-CUX1 #1, respectively.

D  BiFC assay revealing the interaction (arrowheads) of EWSR1 and MAZ in IMR32 and SH-SY5Y cells stably transfected as indicated, with nuclei stained by DAPI. Scale bars: 10 μm.

E  ChIP assay showing the binding of MAZ (normalized to input, *n* = 5) to target gene promoters in IMR32 cells stably transfected as indicated. One-way ANOVA, *\*P* < 0.05 versus mock+sh-Scb.

F  Dual-luciferase assay revealing the relative activity of *p200 CUX1* promoter with wild-type or mutant MAZ-binding site in IMR32 cells stably transfected as indicated (*n* = 5). One-way ANOVA, *\*P* < 0.05 versus mock+sh-Scb.

G, H  Real-time qRT–PCR (G, normalized to β-actin, *n* = 5) and Western blot (H) assays showing the expression of *EWSR1*, *MAZ,* and their target genes in IMR32 cells stably transfected as indicated. One-way ANOVA, *\*P* < 0.05 versus mock+sh-Scb.

Data information: Data are presented as mean ± SEM. Exact *P* values are specified in Appendix Table S4.

*ENO1*, *GPI*, and *PGK1* in NB cells. As a glycolytic enzyme, ENO1 acts as a metabolic tumor promoter by contributing to Warburg effect (Chen *et al*, 2018). GPI is a housekeeping cytosolic enzyme responsible for catalytic interconversion between glucose-6-phosphatase and fructose-6-phosphate, and plays a key role in glycolytic pathway (Ždralević *et al*, 2017). During the glycolytic process, PGK1 contributes to ATP generation and participates in tumor progression (Li *et al*, 2016). Our gain- and loss-of-function studies indicated that *CUX1* promoted the aerobic glycolysis, growth, and invasiveness of NB cells, suggesting its oncogenic roles in NB progression.

Human *CUX1* gene locates at chromosome 7q22, a region associated with copy number gain that contributes to multidrug resistance in NB (Mazzocco *et al*, 2015). However, we found no alteration of copy number or genetic variants of *CUX1* in NB cohorts, indicating other mechanisms facilitating its over-expression. CircRNAs play important roles in regulating gene expression at post-transcriptional or transcriptional levels (Hansen *et al*, 2013; Li *et al*, 2015b). For example, *ciRS-7* and *circSry* serve as sponges of miR-7 and miR-138 in the cytoplasm (Hansen *et al*, 2013). Exonic circRNAs also exert regulatory functions in the cytoplasm by forming a ribonucleoprotein complex with miRNA and AGO protein (Lasda & Parker, 2014). Meanwhile, exon–intron circRNAs are predominantly localized in the nucleus and regulate their parent gene expression in a *cis*-acting manner through specific RNA–protein interaction (Li *et al*, 2015b). In this study, *circ-CUX1* was identified as an intron-containing circRNA up-regulated in tumor tissues and cells. *Circ-CUX1*

enhanced the expression of *CUX1* at transcriptional level, and tumor-promoting functions of *circ-CUX1* were mediated, at least in part, through interacting with EWSR1 protein in NB cells.

As one member of EWS family of RNA-binding proteins, EWSR1 participates in gene transcription, splicing, and miRNA processing (Luo *et al*, 2015). Chromosomal translocation of *ESWR1* has been discovered in Ewing sarcoma (Sohn *et al*, 2010). However, our results revealed no *EWS-Fli1* gene fusion in NB cells. Due to lack of DNA-binding domain, EWSR1 usually acts as a potent transcriptional cofactor in tumor progression via interacting with transcription regulatory proteins, such as CREB-binding protein and p300 (Chakravarti *et al*, 1996). In this study, we found that RRM domain of EWSR1 was necessary for its interaction with ZNF domain of MAZ. As a ubiquitously expressed transcription factor, MAZ binds to GC-rich *cis*-elements through its C2H2-type ZNF motif (Parks & Shenk, 1996) and activates transcription of *KRAS* and vascular endothelial growth factor (*VEGF*) in pancreatic cancer, cervical cancer, and glioblastoma cells (Smits *et al*, 2012; Cogoi *et al*, 2013). Our evidence indicated that through interplay with its cofactor EWSR1, MAZ regulated the transcription of *CUX1*, *S100A9*, *MUC4*, *KLF10*, or *TXNIP* in NB cells. Notably, *circ-CUX1* bound to RRM region of EWSR1, resulting in EWSR1-mediated MAZ transactivation, suggesting the oncogenic roles of *circ-CUX1*/EWSR1/MAZ axis in aerobic glycolysis and tumor progression.

In summary, we demonstrate that elevated *CUX1* and its generated *circ-CUX1* are associated with poor outcome of NB patients,

**Figure 6.  Inhibitory peptides suppress tumor progression by blocking *circ-CUX1*-EWSR1 interaction.**

A, B  Schematic illustration (A) and distribution (arrowheads) (B) of mutant control (CTLP) or inhibitory (EIP-22) peptide within SH-SY5Y cells (at 48 h), with nuclei and cellular membrane stained by DAPI or DiI. Scale bars: 10 μm.

C, D  Western blot (C), RIP, and real-time qRT–PCR (D, *n* = 5) assays indicating the proteins (EWSR1, AMPKβ1) and transcripts (*circ-CUX1*, *pri-miR-222*, *circACC1*) pulled down by biotin-labeled sense (S) or antisense (AS) junction probes of *circ-CUX1*, and antisense probes of *pri-miR-222* or *circACC1* in SH-SY5Y cells treated with CTLP or EIP-22 (15 μM) for 48 h. Student's *t*-test, *\*P* < 0.05 versus IgG.

E, F  Soft agar (E) and Matrigel invasion (F) assays indicating the growth and invasion of SH-SY5Y cells treated with CTLP or EIP-22 (15 μM) for 48 h (*n* = 5). Scale bars: 100 μm. Student's *t*-test, *\*\*P* < 0.01 versus CTLP.

G, H  Representative images, *in vivo* growth curve, tumor weight, Ki-67 and CD31 expression of xenograft tumors (G) and lung metastatic colonization, and Kaplan–Meier curves (H) of nude mice (*n* = 5 for each group) treated with subcutaneous or tail vein injection of SH-SY5Y cells and CTLP or EIP-22 (5 mg kg$^{-1}$). Scale bars: 100 μm. One-way ANOVA, Student's *t*-test, *\*\*P* < 0.01, *\*\*\*P* < 0.001 versus CTLP. Log-rank test for survival comparison.

I  Schematic illustration of *circ-CUX1*-promoted tumor progression: *circ-CUX1* binds to EWSR1 to facilitate its interaction with MAZ, resulting in MAZ transactivation and transcriptional alteration of *CUX1* and other genes associated with aerobic glycolysis and tumor progression. An inhibitory peptide blocking *circ-CUX1*-EWSR1 interaction suppresses tumor progression.

Data information: Data are presented as mean ± SEM. Exact *P* values are specified in Appendix Table S4.

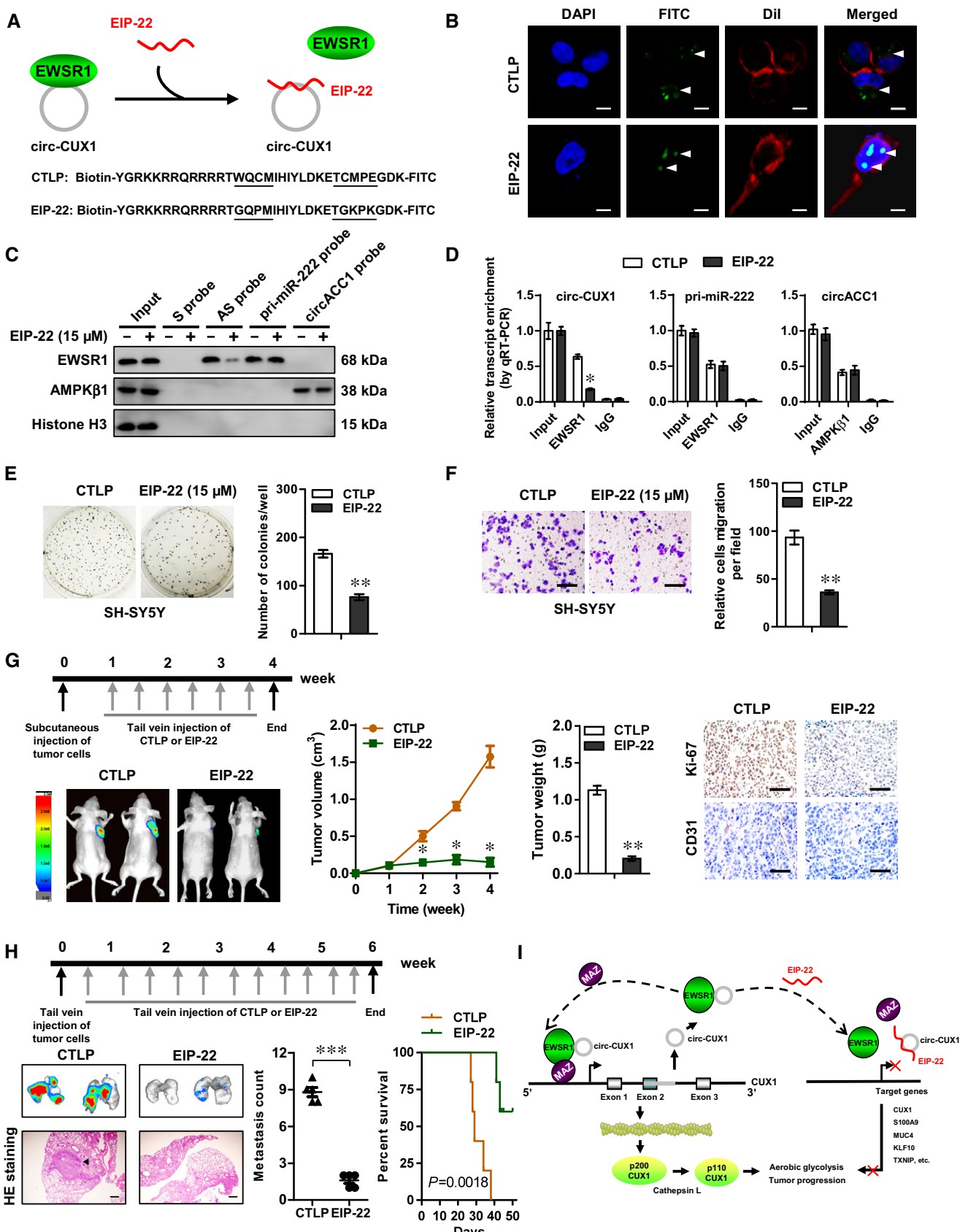

**Figure 6.**

and exert oncogenic roles in aerobic glycolysis and tumor progression. Mechanistically, *circ-CUX1* binds to EWSR1 protein to facilitate MAZ transactivation, resulting in transcriptional alteration of *CUX1* and other genes associated with NB progression. An inhibitory peptide (EIP-22) blocking *circ-CUX1*-EWSR1 interaction or lentivirus-mediated *circ-CUX1* knockdown suppresses the aerobic glycolysis, tumorigenesis, and aggressiveness of NB cells. Combinational administration of EIP-22 and glycolysis inhibitors (2-DG or 3-BP) targeting HK2, GPI, or GAPDH (Cardaci *et al*, 2012; Zhang *et al*, 2014) exerts synergistic effects in suppressing growth and aggressiveness of NB cells. Due to limited size of cohort, the prognostic value of *circ-CUX1* and *CUX1* and their association with *MYCN* amplification in NB warrant further investigation. This study extends our knowledge about the regulation of aerobic glycolysis by transcription factor and its generated circRNA, and suggests that *circ-CUX1*/EWSR1/MAZ axis may be a potential therapeutic target for NB.

# Materials and Methods

## Cell culture

Human MCF 10A (CRL-10317), HeLa (CCL-2), SH-SY5Y (CRL-2266), IMR32 (CCL-127), SK-N-AS (CRL-2137), BE(2)-C (CRL-2268), SK-N-MC (HTB-10), LoVo (CCL-229), PC-3 (CRL-1435), HEK293 (CRL-1573), and HEK293T (CRL-3216) cells were obtained from American Type Culture Collection (Rockville, MD), authenticated by short tandem repeat profiling, and used within 6 months after resuscitation of frozen aliquots. Mycoplasma contamination was regularly examined using Lookout Mycoplasma PCR Detection Kit (Sigma, St. Louis, MO). Tumor cells, HEK293, and HEK293T cells were cultured in RPMI1640 supplied with 10% fetal bovine serum (Life Technologies, Inc., Gaithersburg, MD), while MCF 10A cells were cultured in DMEM/F12 medium containing 5% horse serum (Invitrogen, Carlsbad, CA) and 20 ng ml$^{-1}$ epidermal growth factor (PeproTech, Rocky Hill, NJ) at 37°C, and treated with E64D, actinomycin D (ActD), 2-DG, or 3-BP (Sigma).

## RT–PCR and real-time quantitative RT–PCR

Nuclear, cytoplasmic, and total RNA was extracted using RNA Subcellular Isolation Kit (Active Motif, Carlsbad, CA) or RNeasy Mini Kit (Qiagen Inc., Redwood City, CA), with or without RNase R (3 U μg$^{-1}$, Epicenter, Madison, WI) digestion at 37°C for 15 min. Reverse transcription and real-time PCR were performed using Transcriptor First Strand cDNA Synthesis Kit (Roche, Indianapolis, IN), SYBR Green PCR Master Mix (Applied Biosystems, Foster City, CA), and primers (Appendix Table S1). The transcript levels were analyzed by $2^{-\Delta\Delta Ct}$ method. *De novo* RNA synthesis was blocked by ActD (5 μg ml$^{-1}$) treatment, while mRNA stability was examined by transcript levels at indicated time points.

## Northern blot

The non-junction and junction probes specific for *circ-CUX1* were synthesized and labeled by digoxigenin (DIG, Appendix Table S2). For Northern blot, 20 μg of total RNA was separated on 3-(N-morpholino)propanesulfonic acid-buffered 2% (w/v) agarose gel containing 1.2% (v/v) formaldehyde under denaturing condition at 80 V for 4 h, and transferred to Hybond-N+ membrane (Pall Corp., Port Washington, NY). Hybridization was performed at 65°C for 16–18 h in DIG Easy Hyb solution (Roche) and detected by anti-DIG antibody (1:500 dilution) and chemiluminescence substrate CSPD (Roche).

## Western blot

Tissue or cellular protein was extracted with 1× cell lysis buffer (Promega, Madison, WI). Western blot was performed as previously described (Zhang *et al*, 2012; Zhao *et al*, 2016; Li *et al*, 2018b), with antibodies (1:500 dilution) specific for CUX1 (sc-514008, Santa Cruz Biotechnology, Santa Cruz, CA), ENO1 (ab155102), GPI (ab66340), PGK1 (ab38007), EWSR1 (ab93837), ELAVL1 (ab136542), SYNCRIP (ab184946), MAZ (ab85725), MUC4 (ab60720), S100A9 (ab92507), KLF10 (ab73537), TXNIP (ab188865), AMPKβ1 (ab32112), β-actin (ab8227), Flag (ab18230), Myc (ab9106, Abcam Inc., Cambridge, MA), GST (sc-33614), or histone H3 (sc-10809, Santa Cruz Biotechnology).

## Gene over-expression and knockdown

Human *circ-CUX1* linear sequence (393 bp) was obtained from NB tissues by PCR (Appendix Table S2) and inserted into pLCDH-ciR (Geenseed Biotech Co., Guangzhou, China). Human *p200 CUX1* construct was provided by Dr. George Stratigopoulos, while *p110 CUX1* was released by digestion or amplified using primers (Appendix Table S2), and subcloned into pcDNA3.1 (Invitrogen) or pCMV-3Tag-1A (Addgene, Cambridge, MA). Human *EWSR1* cDNA (1,971 bp) and *MAZ* cDNA (1,482 bp) were provided by Dr. Ralf Janknecht or amplified from NB tissues with primers (Appendix Table S2), and their truncations were subcloned into pCMV-N-Myc or pCMV-3Tag-1A (Addgene). Mutation of *circ-CUX1* or *EWSR1* was prepared with GeneTailor™ Site-Directed Mutagenesis System (Invitrogen) and primers (Appendix Table S2). Oligonucleotides specific for shRNAs (Appendix Table S3) were inserted into GV298 (GeneChem Co., Ltd, Shanghai, China). Stable cells were screened by neomycin or puromycin (Invitrogen).

## Rescue of target gene expression

To rescue *circ-CUX1* knockdown-altered target gene expression, *EWSR1* or *MAZ* was transfected into stable cell lines. To restore target gene expression altered by *circ-CUX1*, shRNAs against *EWSR1* or *MAZ* (Appendix Table S3) were transfected into tumor cells using GeneSilencer Transfection Reagent (Genlantis, San Diego, CA).

## Lentivirus packaging

Lentiviral vectors were co-transfected with packaging plasmids psPAX2 and pMD2G (Addgene) into HEK293T cells. Infectious lentivirus was harvested at 36 and 60 h after transfection, and filtered through 0.45 μm PVDF filters. Recombinant lentivirus was concentrated 100-fold by ultracentrifugation (2 h at 120,000 *g*), dissolved in phosphate-buffered saline (PBS), and injected into mice within 48 h.

## RNA-seq assay

Total RNA of tumor cells ($1 \times 10^6$) was extracted using TRIzol® reagent (Life Technologies, Inc.). Library preparation and transcriptome sequencing on an Illumina HiSeq X Ten platform were carried out at Novogene Bioinformatics Technology Co., Ltd. (Beijing, China). Fragments per kilobase of transcript per million fragments mapped (FPKM) of each gene were calculated.

## Dual-luciferase reporter assay

Complementary oligonucleotides containing four canonical binding sites of CUX1 or MAZ, and promoter fragments of *ENO1* (−1,880/+301), *GPI* (−1,854/+247), *PGK1* (−882/+246), or *p200 CUX1* (−2,084/+106) amplified from genomic DNA (Appendix Table S2) were subcloned into pGL3-Basic (Promega). Mutation of MAZ-binding site was performed with GeneTailor™ Site-Directed Mutagenesis System (Invitrogen) and primers (Appendix Table S2). To test specificity of shRNA, target sequences of *circ-CUX1* and *CUX1* were amplified using primers (Appendix Table S2) and subcloned into 3′-untranslated region of *Renilla* luciferase within psiCHECK2 (Promega). Dual-luciferase assay was performed as previously described (Zhang *et al*, 2012; Zhao *et al*, 2016; Li *et al*, 2018b).

## RNA pull-down and mass spectrometry

Biotin-labeled oligonucleotide probes targeting junction sites of circRNAs were synthesized (Invitrogen). Linear *circ-CUX1* was *in vitro* transcribed using Biotin RNA Labeling Mix (Roche) and T7 RNA polymerase, incubated with guide oligonucleotides (Appendix Table S2), circularized using T4 RNA ligase I, treated with RNase R, and purified with RNeasy Mini Kit (Qiagen Inc.). RNA pull-down was performed as previously described (Li *et al*, 2018b). Retrieved protein was detected by Western blot or mass spectrometry (Wuhan Institute of Biotechnology, Wuhan, China), while recovered transcripts were measured by RT–PCR using primers (Appendix Table S1). In ChIRP assay, cells were harvested, cross-linked, sonicated, hybridized with probes, and mixed with streptavidin magnetic beads (Chu & Chang, 2016). The retrieved DNA was detected by PCR using primers (Appendix Table S1).

## RNA-FISH

Biotin-labeled antisense or sense probe for *circ-CUX1* junction was synthesized (Appendix Table S2). The probes for *GAPDH* and *U1* were generated by *in vitro* transcription of PCR products (Appendix Table S1) using DIG Labeling Kit (MyLab Corporation, Beijing, China). Cells were incubated with 40 nM FISH probe in hybridization buffer (100 mg ml$^{-1}$ dextran sulfate, 10% formamide in $2 \times$ SSC) at 37°C for 16 h, with or without RNase R (3 U µg$^{-1}$) treatment. The signals of *circ-CUX1* were detected by Fluorescent *In Situ* Hybridization Kit (RiboBio, Guangzhou, China), with nuclei staining by 4′,6-diamidino-2-phenylindole (DAPI).

## Fluorescence immunocytochemical staining

Tumor cells were grown on coverslips, incubated with antibodies specific for EWSR1 (ab93837; Abcam Inc.; 1:100 dilution) at 4°C overnight, and treated with Alexa Fluor 594 goat anti-rabbit IgG (1:1,000 dilution) and DAPI (300 nM) staining. The images were photographed under a Nikon A1Si Laser Scanning Confocal Microscope and applied for 3D reconstruction using NIS-Elements Viewer (Nikon Instruments Inc., Japan).

## Co-immunoprecipitation (co-IP)

Co-IP was performed as previously described (Jiao *et al*, 2018; Li *et al*, 2018b), with antibodies (1:200 dilution) specific for EWSR1 (sc-28327, Santa Cruz Biotechnology), MAZ (ab85725), Flag (ab18230), or Myc (ab9106, Abcam Inc.). Bead-bound proteins were released and analyzed by Western blot.

## Bimolecular fluorescence complementation system (BiFC)

Human *EWSR1* cDNA (1,971 bp) and *MAZ* cDNA (1,482 bp) were subcloned into BiFC vectors pBiFC-VN173 and pBiFC-VC155 (Addgene), and co-transfected into tumor cells for 24 h. The fluorescence emission was observed under a confocal microscope, with excitation and emission wavelengths of 488 and 500 nm, respectively (Kerppola, 2008).

## Chromatin immunoprecipitation (ChIP)

ChIP assay was undertaken using EZ-ChIP kit (Upstate Biotechnology, Temecula, CA) (Zhao *et al*, 2016; Li *et al*, 2018b), with antibodies (1:100 dilution) specific for CUX1 (#81557, Cell Signaling Technology, Inc., Danvers, MA) or MAZ (ab85725, Abcam Inc.) and primers targeting gene promoters (Appendix Table S1).

## Cross-linking RNA immunoprecipitation (RIP)

Cells ($1 \times 10^8$) were ultraviolet light cross-linked at 254 nm (200 J cm$^{-2}$). RIP assay was performed using Magna RIP™ RNA-Binding Protein Immunoprecipitation Kit (Millipore) (Zhao *et al*, 2016; Li *et al*, 2018b), with antibodies (1:100 dilution) specific for AGO2 (ab186733, Abcam Inc.), EWSR1 (#11910, Cell Signaling Technology, Inc.), GST (sc-33614, Santa Cruz Biotechnology), or AMPKβ1 (ab32112, Abcam Inc.). Co-precipitated RNAs were detected by RT–PCR or real-time qRT–PCR with specific primers (Appendix Table S1).

## *In vitro* binding assay

A series of *EWSR1* truncations were amplified with primers (Appendix Table S2), subcloned into pGEX-6P-1 (Addgene), and transformed into *E. coli* to produce GST-tagged EWSR1 protein (Zhao *et al*, 2016; Li *et al*, 2018b). The EWSR1-circRNA complexes were pulled down using GST beads (Sigma). Protein was detected by SDS–PAGE and Western blot, while circRNA was measured by RT–PCR with specific primers (Appendix Table S1).

## RNA EMSA

Biotin-labeled *circ-CUX1* probe was synthesized as described above. RNA EMSA using nuclear extracts was performed using LightShift Chemiluminescent RNA EMSA Kit (Thermo Fisher Scientific, Inc., Waltham, MA).

**Design and synthesis of inhibitory peptides**

Based on interacting region of EWSR1 revealed by mutagenesis and *in vitro* binding assays, wild-type and mutant inhibitory peptides blocking *circ-CUX1* and EWSR1 interaction were designed and synthesized by linking with biotin-labeled 11 amino acid cell-penetrating peptide (YGRKKRRQRRR) from Tat protein transduction domain at the N-terminus and conjugating with fluorescein isothiocyanate (FITC) at the C-terminus (ChinaPeptides Co. Ltd, Shanghai, China), with purity larger than 95%.

**Biotin-labeled peptide pull-down**

Total RNA was isolated using RNeasy Mini Kit (Qiagen Inc.) and incubated with biotin-labeled peptide at 4°C overnight. Then, incubation of RNA-peptide complex with streptavidin-agarose was undertaken at 4°C for 2 h. Beads were extensively washed, and circRNAs pulled down were measured by real-time qRT–PCR.

**Aerobic glycolysis and seahorse extracellular flux assays**

Cellular glucose uptake, lactate production, and ATP levels were detected as previously described (Ma *et al*, 2014). ECAR and OCR were measured in XF media under basal conditions and in response to glucose (10 mM), oligomycin (2 μM), and 2-deoxyglucose (50 mM) using a Seahorse Biosciences XFe24 Flux Analyzer (North Billerica, MA).

***In vitro* cell viability, growth, and invasion assays**

The 2-(4,5-dimethylthiazol-2-yl)-2,5-diphenyl tetrazolium bromide (MTT; Sigma) colorimetric (Li *et al*, 2015a, 2018b), soft agar (Zhang *et al*, 2012; Zhao *et al*, 2016; Li *et al*, 2018a,b), and Matrigel invasion (Zhang *et al*, 2012; Zhao *et al*, 2016; Li *et al*, 2018a,b) assays were undertaken to measure *in vitro* viability, growth, and invasive capabilities of tumor cells.

***In vivo* growth, metastasis, and therapeutic assays**

All animal experiments were carried out in accordance with NIH Guidelines for the Care and Use of Laboratory Animals, and approved by the Animal Care Committee of Tongji Medical College (approval number: Y20080290). For *in vivo* tumor growth and experimental metastasis studies, tumor cells ($1 \times 10^6$ or $0.4 \times 10^6$) were injected into dorsal flanks or tail vein of blindly randomized 4-week-old female BALB/c nude mice (National Rodent Seeds Center, Shanghai, China) breeding at specific pathogen free (SPF) condition ($n = 5$ per group) (Zhang *et al*, 2012; Zhao *et al*, 2016; Li *et al*, 2018a,b). For *in vivo* therapeutic studies, tumor cells ($1 \times 10^6$ or $0.4 \times 10^6$) were injected into dorsal flanks or tail vein of nude mice, respectively. One week later, mice were blindly randomized and treated by tail vein injection of synthesized cell-penetrating peptide (ChinaPeptides, Shanghai, China) or lentivirus ($1 \times 10^7$ plaque-forming units) as indicated. The *in Vivo* Optical Imaging System (*In-Vivo* FX PRO, Bruker Corporation, Billerica, MA) was applied to acquire fluorescent images of xenograft tumors in nude mice.

**The paper explained**

**Problem**
Neuroblastoma (NB) is the most common extracranial tumor in childhood. Although countless efforts have been made to improve the therapeutic efficiency, the outcome of patients suffering from high-risk NB still remains poor. Aerobic glycolysis is a hallmark of metabolic reprogramming that contributes to tumor progression. Further in-depth investigation of mechanisms regulating aerobic glycolysis during NB progression is vital for resolution of these issues.

**Results**
By integrating analysis of public datasets, we identify that *CUX1* and *CUX1*-generated circular RNA (*circ-CUX1*) facilitate aerobic glycolysis and NB progression. Mechanistically, transcription factor p110 CUX1, a proteolytic product of p200 CUX1, promotes expression of glycolytic genes, while *circ-CUX1* facilitates EWSR1-mediated MAZ transactivation to alter expression of *p200 CUX1* and other genes associated with tumor progression. High expression of *circ-CUX1* or *p200 CUX1* is associated with poor outcome of NB patients. Administration of an inhibitory peptide blocking *circ-CUX1*-EWSR1 interaction or lentivirus mediating *circ-CUX1* knockdown suppresses aerobic glycolysis, growth, and aggressiveness of NB cells.

**Impact**
Our results extend the knowledge about regulation of aerobic glycolysis by transcription factor and its generated circRNA. First of all, transcription factor CUX1 is essential for glycolytic gene expression during NB progression. Secondly, EWSR1 interacts with MAZ to facilitate its transactivation and regulate downstream gene expression. Thirdly, *circ-CUX1* facilitates the interaction of EWSR1 with MAZ. Finally, blocking *circ-CUX1*-EWSR1 interaction might be a novel therapeutic strategy for NB and other tumors.

**Patient tissue samples**

Human tissue study was approved by the Institutional Review Board of Tongji Medical College (approval number: 2011-S085). All procedures were conformed to principles set forth by Declaration of Helsinki and Department of Health and Human Services Belmont Report. Written informed consent was obtained from all patients without preoperative chemotherapy or other treatment. Fresh tumor tissues were collected at surgery, validated by pathological diagnosis, and stored at −80°C. Total RNAs of normal fetal adrenal medulla were purchased from Clontech (Mountain View, CA).

**Immunohistochemistry**

Immunohistochemical staining and quantitative evaluation were performed as previously described (Zhang *et al*, 2012; Zhao *et al*, 2016; Li *et al*, 2018a,b), with antibodies specific for Ki-67 (1:500, sc-23900, Santa Cruz Biotechnology) or CD31 (1:500, ARG52748, Arigo, Hsinchu City, Taiwan). The degree of positivity was blindly assessed by at least two pathologists.

**Statistical analysis**

All data were shown as mean ± standard error of the mean (SEM). Cutoff of gene expression was defined by average values. Two-sided unpaired Student's *t*-test and one-way ANOVA were used to

compare difference. Pearson's correlation coefficient assay was used to analyze expression correlation. Log-rank test and Cox regression models were used to assess survival difference and hazard ratio. All statistical tests were considered significant when $P < 0.05$. Randomization and blinding strategies were used whenever possible. Experimental sample size was determined on the basis of power analysis assuming a significance level (alpha) of 0.05 and a power of 80%. Animal cohort sizes were determined on the basis of similar studies. The exact $P$-values and number of replicates were indicated in Appendix Table S4.

## Data availability

RNA-seq data have been deposited in Gene Expression Omnibus (GEO) repository (https://www.ncbi.nlm.nih.gov/geo), under accession number GSE136135.

**Expanded View** for this article is available online.

## Acknowledgements

We appreciate Drs. George Stratigopoulos and Ralf Janknecht for providing vectors. This work was granted by the National Natural Science Foundation of China (Grants # 81472363, 81572423, 81672500, 81773094, 81772967, 81874085, 81874066, 81802925, 81903011, 81903008), and Fundamental Research Funds for the Central Universities (Grant # 2019kfyRCPY032).

## Author contributions

Conception and design: QT and LZ; Methodology and resources: HL, FY, XW, EF, and DL; Acquisition of data: HL, FY, AH, XW, EF, YC, DL, HS, JW, YG, YL, and HJL; Supervision: QT, LZ, and KH; Writing the manuscript: HL, FY, QT, and LZ.

## Conflict of interest

The authors declare that they have no conflict of interest.

## For more information

Publically available datasets can be found
(i)   GEO database (GSE16476, GSE62564, GSE41258, GSE5851, GSE6956, https://www.ncbi.nlm.nih.gov/geo).
(ii)  The Cancer Genome Atlas (TCGA, https://portal.gdc.cancer.gov).
(iii) Kaplan–Meier plotter (http://kmplot.com).
(iv)  Oncogenomics (https://pob.abcc.ncifcrf.gov/cgi-bin/JK).
(v)   cBioPortal for Cancer Genomics (http://cbioportal.org).
(vi)  Tumor alterations relevant for genomics-driven therapy (TARGET, https://software.broadinstitute.org/cancer/cga/target).

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
