## [Review Process File · EMBO Molecular Medicine]

Therapeutic targeting of circ-CUX1/EWSR1/MAZ axis inhibits glycolysis and neuroblastoma progression

Huanhuan Li, Feng Yang, Anpei Hu, Xiaojing Wang, Erhu Fang, Yajun Chen, Dan Li, Huajie Song, Jianqun Wang, Yanhua Guo, Yang Liu, Hongjun Li, Kai Huang, Liduan Zheng, Qiangsong Tong

Review timeline:	Submission date:	2 May 2019
	Editorial Decision:	4 June 2019
	Revision received:	4 September 2019
	Editorial Decision:	14 October 2019
	Revision received:	15 October 2019
	Accepted:	18 October 2019

Editor: Jingyi Hou

Transaction Report:

1st Editorial Decision

4 June 2019

Thank you for the submission of your manuscript to EMBO Molecular Medicine. We have now heard back from three referees who evaluated your manuscript.

As you will see from the reports below, the three Referees mention the potential interest and clinical relevance of the study. However, they also raise substantial concerns about your work, which should be convincingly addressed in a major revision of the present manuscript. In particular, additional controls and experiments are required to confirm the interaction and co-localization of cir-CUX1 (rather than its linear form) and its associated proteins. Further, comments about cir-CUX1 overexpression and knockdown specificity should be addressed, along with other technical concerns as recommended by Referee #1.

We would welcome the submission of a revised version within three months for further consideration and would like to encourage you to address all the criticisms raised as suggested to improve conclusiveness and clarity. Please note that EMBO Molecular Medicine strongly supports a single round of revision and that, as acceptance or rejection of the manuscript will depend on another round of review, your responses should be as complete as possible.

I look forward to receiving your revised manuscript.

***** Reviewer's comments *****

Referee #1 (Comments on Novelty/Model System for Author):

The model systems utilized are adequate to test the hypothesis raised.

Referee #1 (Remarks for Author):

In the manuscript entitled "Therapeutic targeting of circ-CUX1/EWSR1/MAZ axis inhibits glycolysis and neuroblastoma progression", Li et al. describe a circuit for the regulation of glycolysis in cancer, in particular in neuroblastoma cells. This circuit involves a circular RNA able to sustain its host gene expression, therefore activating glycolytic metabolism and promoting cancer progression. This work is potentially interesting, especially for the therapeutic implications proposed in its last section. However, even if the work could be worth being published in principle, there are several important issues and technical points which need further analysis and clarification.

Major comments

-S1A: "In NB cell lines, cervical cancer HeLa cells, colon cancer LoVo cells, and prostate cancer PC-3 cells, but not in non-transformed normal MCF 10A cells, p200 and p110 were the major CUX1 isoforms, while p75 (Goulet et al, 2002) and CDP/cut alternatively spliced cDNA (CASP) (Gillingham et al, 2002) were undetectable or expressed at low levels (Appendix Fig S1A)".

In the western blot in S1A, CASP seems to be more expressed in cancer cells than in MCF cells. Its levels in cancer cells are very similar to p110: is it a matter of different membrane exposures? In IMR32 cells, it also seems that p110 is the less represented isoform. Actually, while it is clear that the CASP RNA isoform is not differentially expressed between normal and cancer cells, it is less clear the difference in its protein levels. Moreover, the authors should explain why it is important to analyze the relative ratio between the CUX1 different isoforms, for the purposes of the paper.

- S1F: A western blot showing CASP efficient knock-down and ENO1, GPI and PGK protein levels should be shown.

- Fig. 1D-1E: "The CUX1 enrichment and promoter activity of ENO1, GPI, and PGK1 were increased and decreased by over-expression or knockdown of p110 CUX1 in NB cells, respectively (Fig 1D and Fig 1E)".

If the antibody used for the p100-ChIP assay recognizes also p200 (as described in S1A), the authors can overexpress a tagged p100 protein, and use a tag-specific antibody for the assays, to be sure to focus only on p110 isoform.

Moreover, the authors describe a "p110 knock-down" in the text, but the shRNAs used (shCUX1 #1-#2) target only the p200 RNA, according to S1A. To be sure that the effect seen is not related to the uncut p200, maybe they can perform the same assays in presence of E64D inhibitor: this should be equivalent to a "specific" p110 knock-down. This should be taken into account also for the other experiments presented in the work.

- Fig. 2 G-H (and S5A-B-C-D): in order to prove the specificity of sh-circ-CUX1#1-#2, the authors should demonstrate that they have no effect on the linear mRNA. This is very important also to conclusively demonstrate that the effects on cancer cell proliferation in vitro and in vivo are circRNA-dependent. In the overexpression experiments the nuclear localization of the exogenous circ-CUX1 should be shown. The plasmid generating the exogenous circ-CUX1 is likely to produce also quite high levels of linear unspliced precursor. How the authors can be sure that the observed effects are not due to this transcript?

The statement that circ-CUX1 exerts its role at the transcriptional level should be more clearly demonstrated. The authors should demonstrate that the activity is not elicited at level of CUX1 mRNA stability (analysis of CUX1 mRNA levels upon actinomycin treatment); moreover, they should check whether circ-CUX1 localizes, according to the model, at the sites of the p200 CUX1 promoter or of its target genes' transcription (ChIRP assay or double RNA/DNA FISH).

- Fig. 4A: in order to have reliable results about circ-CUX1 associated proteins, a native or crosslinked (preferred in order to exclude spurious interactions in solution) pulldown of endogenous circ-CUX1 should be performed. Moreover, the inverse experiment (RIP or CLIP) should confirm

the interaction also in non-transfected cells. The pulldown should also be specific, or at least enriched for the circular form of CUX1, rather than the linear CUX1 mRNA. Therefore, the levels of both the circRNA and the mRNA should be checked in each pulldown or IP.

- Fig. 4C: it is very difficult to conclude about the colocalization of circ-CUX1 and the protein. In case a 3D-confocal reconstructions should be provided. The image is quite surprising in terms of number of circ-CUX1 molecules, it is not clear from the legends whether it refers to an OE experiment. If the image corresponds to endogenous circ-CUX1 levels, they are very high. Have the authors checked what is the abundance of the endogenous RNA? Considering the levels of circ-CUX1, and in order to prove the model, a ChIRP assay or double DNA/RNA FISH should be provided.

-Fig. 6C-D: in order to appreciate the effect of the synthetic peptide on MAZ transactivation and S100A9, MUC4, KLF10 and TXNIP proteins, a western blot similar to the ones in Fig. 5H and S10C should be shown. Have the authors tested the specificity of the effect of the peptide on other RNP complexes?

Minor comments

Introduction - page 3: "For example, circRNA antisense to cerebellar-degeneration-related protein 1 (CDR1as) harbors 63 conserved miRNA binding sites for miR-7".

Nota that actually, in Piwecka et al., 2017 the model of CDR1as as a sponge has been revised. This point should be discussed appropriately.

Cell lines: the authors should specify in the text and figure legends which are the "NB cell lines" used in the work, to better understand the results.

S1B: "The transcript levels of p200 CUX1, but not of CASP, were higher than those in normal fetal adrenal medulla ($P < 0.05$, Appendix Fig S1A), especially in NB cases with poor stroma ($P < 0.0001$) or advanced INSS stages ($P = 0.0081$), without association with MYCN amplification ($P = 0.3511$, Appendix Fig S1B)".

From the boxplot, it seems that CUX1 is higher in cases with MYCN amplification ("Yes" column). Same comment for S4F.

Fig. 2B: gel showing circ-CUX1 amplification should not be cropped, in order to see concatemers generated by the amplification of a circRNA-derived product of reverse transcription.

Fig. 2D: is it sure that the RNaseR was used 3 U μ g⁻¹ and not 30 U μ g⁻¹ of RNA?

Fig. 4: there is a problem in how the panels are labeled in the figure and in the text.

Referee #2 (Remarks for Author):

In this manuscript, Li, H. et al reports that a circular RNA that encodes p200 CUT-like hemeobox 1 (CUX1) regulates neuroblastoma (NB) tumorigenicity through modulation of aerobic glycolysis. By using bioinformatic analysis of public datasets, molecular, cell biology and biochemical assays, the authors showed that the proteolytic products of p200 CUX1, p110 CUX1, a transcription factor induced genes of key enzymes in glycolysis pathway. The circular CUX1 RNA (circ-CUX1) binds to EWS RNA binding protein 1 (EWSR) that enhances EWSR interaction with a Myc-associated zinc finger protein MAZ, leading transactivation of MAZ and stimulation of gene expression of p200 CUX1 and other genes, promoting NB tumorigenicity. Inhibition of CUX1 expression or disruption of circ-CUX1-EWSR interaction inhibited NB tumor aerobic glycolysis, cell proliferation, NB tumor growth and metastasis. Lastly, expression of circ-CUX1 is associated with prognosis of patients with NB. Overall, this is an excellent study with high significance. The study is well designed and executed. The data strongly support the hypothesis and overall conclusion with high quality. There are few minor weaknesses, however. If the authors are able to address these concerns, this study will be further strengthened.

Comments:

1) There are many bar graphs throughout the entire data presented in this manuscript. However, it is very difficult to distinguish the differences among veracious experimental groups due to the current drawing. The authors must re-scale most if not all bar graphs, in particular the y-axis to allow readers to clearly see the differences. For example, in Figure 1D, 1E, 1F, and 1G the full scale of Y-axis should be 4.0 or 80 (1F) instead of current ones. The revised bar graphs will allow one to observe the effects of shCUX1 comparing with controls. The labels can be moved or placed on other

places instead of at the top portion of the bar graphs. Similar revisions should also be made in bar graphs in other figures.

2) On page 11, the description of data in Figure 6C is an overstatement. Data in Figure 6C showed that treatment of EIP-22 peptide did not "abolish" but reduced the interaction of circ-CUX1 with EWSR1.

3) In Figure S2B, the authors should also analyze the relationships of genes expression of these four genes with patient survival using TCGA or another well annotated and curated public dataset. Similar concerns also go to data in Figure S4E and S4F. The clinical cases of NB tumors are too low to draw the conclusion. For example, in S4E, the last graph, one box only has a total of 9 NB samples. This number is too small to draw statistical conclusion.

Referee #3 (Remarks for Author):

The work of Li H., and colleagues demonstrates in a very original manner the role of CUX1 in neuroblastoma. Authors provide sufficient proves on the tumor promoting role of CUX1 and circ-CUX1. High levels of circ-CUX1 facilitates glycolysis and increased the aggressiveness of NB cells. Mechanistically this is due to the binding with EWSR1 and activation of MAZ followed by transcriptional alteration of genes associated with tumor progression. Taken together the results point at circ-CUX1/EWSR1/MAZ axis as a therapeutic target NB. Interestingly CUX1 seems to be a prognostic factor in NB as well as in other cancer types.

Overall the experimental strategies used by the authors are adequate and the results are convincing and original.

A few minor comments:

1-Does the combination of glycolysis inhibitors and anti circ-CUX1 targeting synergize against NB tumor growth?

2-Are the changes in ATP levels observed deriving only from glycolysis? Would be interesting to demonstrate that there are not disfunctions.

1st Revision - authors' response

4 September 2019

Referee #1

1. S1A: "In NB cell lines, cervical cancer HeLa cells, colon cancer LoVo cells, and prostate cancer PC-3 cells, but not in non-transformed normal MCF 10A cells, p200 and p110 were the major CUX1 isoforms, while p75 (Goulet et al, 2002) and CDP/cut alternatively spliced cDNA (CASP) (Gillingham et al, 2002) were undetectable or expressed at low levels (Appendix Fig S1A)". In the western blot in S1A, CASP seems to be more expressed in cancer cells than in MCF cells. Its levels in cancer cells are very similar to p110: is it a matter of different membrane exposures? In IMR32 cells, it also seems that p110 is the less represented isoform. Actually, while it is clear that the CASP RNA isoform is not differentially expressed between normal and cancer cells, it is less clear the difference in its protein levels. Moreover, the authors should explain why it is important to analyze the relative ratio between the CUX1 different isoforms, for the purposes of the paper.

Response: We appreciate the reviewer's positive comments and revision guidance on our manuscript. In this study, we performed comprehensive analysis of a public dataset of 88 neuroblastoma (NB) cases, and identified CUT-like homeobox 1 (CUX1) as a transcription factor essential for expression of glycolytic genes. Since previous studies have revealed several isoforms of CUX1 (Mol Cell, 2004 14: 207-219; Mol Biol Cell, 2002, 13: 3761-3774; Cancer Res, 2002, 62: 6625-6633), it is rational to analyze their relative levels in NB by real-time quantitative RT-PCR (qRT-PCR) and western blot assays. We are sorry for the confusing description and results in Appendix Fig S1A, mainly due to different membrane exposure in western blot. To resolve this problem, we have repeated the measurement and found that higher transcript levels of CUX1 isoform p200 were noted in NB cell lines, while p75 (Mol Cell, 2004, 14: 207-219) was expressed at very low levels (Appendix Fig S1A). Consistently, elevated levels of p200 CUX1 and its proteolytically processed isoform p110 were noted in NB cell lines, cervical cancer HeLa cells, colon cancer LoVo cells, and prostate cancer PC-3 cells, than those of non-transformed normal MCF 10A cells (Appendix Fig S1A). However, both transcript and protein levels of CDP/cut alternatively spliced cDNA (CASP) (Mol Biol Cell, 2002, 13: 3761-3774) were not differently expressed between normal and tumor cells (Appendix Fig S1A). Thus, we focused

on the roles of p200 and p110 CUX1 in regulating glycolytic gene expression. In this revised manuscript, we have updated results from repeated experiments into Appendix Fig S1A and described these clearly at pages 4-5.

2. S1F: A western blot showing CASP efficient knock-down and ENO1, GPI and PGK protein levels should be shown.

Response: We definitively agree with the reviewer that it is necessary to observe the effects of *CASP* knockdown on expression of enolase 1 (*ENO1*), glucose-6-phosphate isomerase (*GPI*), and phosphoglycerate kinase 1 (*PGK1*) in NB cells. To meet this end, we performed the real-time qRT-PCR and western blot assays. The results indicated that transfection of two short hairpin RNAs (shRNAs) targeting *CASP* resulted in its down-regulation, without significant impact on the transcript and protein levels of *ENO1*, *GPI*, and *PGK1* in SH-SY5Y cells (Appendix Fig S1F and G). In this revised manuscript, we have added the data in Appendix Fig S1G and described these clearly at page 5.

3. Fig. 1D-1E: "The CUX1 enrichment and promoter activity of ENO1, GPI, and PGK1 were increased and decreased by over-expression or knockdown of p110 CUX1 in NB cells, respectively (Fig 1D and Fig 1E)". If the antibody used for the p110-ChIP assay recognizes also p200 (as described in S1A), the authors can overexpress a tagged p110 protein, and use a tag-specific antibody for the assays, to be sure to focus only on p110 isoform.

Response: Good comments. In this study, we investigated the effects of CUX1 on transcription of target genes *ENO1*, *GPI*, and *PGK1* in NB cells by ChIP-qPCR and dual-luciferase assays. Since p110 is an isoform processed from p200 CUX1, we definitively agree with the reviewer that it is necessary to observe the specific roles of p110 CUX1 in these processes. To meet this end, we established a Flag-tagged p110 CUX1 expression vector (Appendix Table S2) for transfection into NB cell lines, and applied the Flag antibody in ChIP assay. We found that the CUX1 enrichment and promoter activity of *ENO1*, *GPI*, and *PGK1* were increased by Flag-tagged p110 CUX1 over-expression in SH-SY5Y and IMR32 cells (Appendix Fig S1H, Fig 1D and E). In this revised manuscript, we have added the data in Appendix Fig S1H, Fig 1D and E, and described these clearly at pages 5 and 33.

4. Moreover, the authors describe a "p110 knock-down" in the text, but the shRNAs used (shCUX1 #1-#2) target only the p200 RNA, according to S1A. To be sure that the effect seen is not related to the uncut p200, maybe they can perform the same assays in presence of E64D inhibitor: this should be equivalent to a "specific" p110 knock-down. This should be taken into account also for the other experiments presented in the work.

Response: In this study, we observed the effects of *CUX1* knockdown on downstream gene expression, aerobic glycolysis, growth, and aggressiveness of NB cells. Since p200 CUX1 protein is proteolytically processed by cathepsin L to generate p110 isoform, shRNAs were designed against *p200 CUX1* and should not be described as "*p110* knockdown". To further investigate the effects of p110 CUX1 on these features of tumor cells, we definitively agree with the reviewer that it is necessary to perform these studies by using inhibitor (E64D) of cathepsin L, an established enzyme for proteolytic processing of p200 CUX1 into p110 isoform (Mol Cell, 2004, 14: 207-219). Our finding revealed that E64D treatment decreased the CUX1 enrichment and promoter activity of *ENO1*, *GPI*, and *PGK1* (Fig 1D and Fig 1E), attenuated the glycolytic process (Fig 1F and Fig 1G), and decreased the glucose uptake, lactate production, ATP levels, anchorage-independent growth, and invasion of IMR32 cells (Appendix Fig S2A-D), which was similar to those of *p200 CUX1* knockdown (Fig. 1D-G and Appendix Fig S2A-D). Since our results also indicated that E64D treatment abolished the elevated levels of *ENO1*, *GPI*, and *PGK1* induced by ectopic expression of *p200 CUX1* (Appendix Fig S1E), we believe that p110 CUX1 plays essential roles in aerobic glycolysis and NB progression. In this revised manuscript, we have added the data into Fig 1D-G and Appendix Fig S2A-D, and described these clearly at page 5.

5. Fig. 2 G-H (and S5A-B-C-D): in order to prove the specificity of sh-circ-CUX1#1-#2, the authors should demonstrate that they have no effect on the linear mRNA. This is very important also to conclusively demonstrate that the effects on cancer cell proliferation in vitro and in vivo are circRNA-dependent. In the overexpression experiments the nuclear localization of the exogenous circ-CUX1 should be shown. The plasmid generating the exogenous circ-CUX1 is likely to produce also quite high levels of linear unspliced precursor. How the authors can be sure that the observed effects are not due to this transcript?

Response: Good comments. In this study, we applied two shRNAs targeting junction site of *circ-CUX1* (sh-*circ-CUX1*). We definitively agree with the reviewer that it is necessary to demonstrate their specificity. To meet this end, we performed RNA immunoprecipitation (RIP) assay to observe the enrichment of argonaute 2 (AGO2) on *circ-CUX1* or *CUX1* mRNA in tumor cells stably transfected with sh-*circ-CUX1*. The results demonstrated that transfection of sh-*circ-CUX1* #1 and sh-*circ-CUX1* #2 resulted in increased AGO2 enrichment on *circ-CUX1*, but not on *CUX1* mRNA, in IMR32, SH-SY5Y, LoVo, and PC-3 cells (Appendix Fig S5B). In addition, a luciferase reporter-based assay monitoring specificity of shRNAs (*Nucleic Acids Res*, 2010, 38: 5761-5773) was applied for further validation, in which target sequences of *circ-CUX1* and *CUX1* were subcloned into 3'-untranslated region of *Renilla* luciferase within psiCHECK2 vector (Promega). We found that transfection of sh-*circ-CUX1* #1 or sh-*circ-CUX1* #2 decreased the luciferase activity of *circ-CUX1* reporter in IMR32, SH-SY5Y, LoVo, and PC-3 cells, without impact on that of *CUX1* reporter (Appendix Fig S5C). These results indicated that shRNAs were established to specifically target *circ-CUX1*, but not *p200 CUX1*, in tumor cells. In this revised manuscript, we have added the data in Appendix Fig S5B and C, and described these clearly at page 7.

We definitively agree with the reviewer that it is necessary to further investigate the localization and effects of *circ-CUX1* in the over-expression experiments. We found that in subcellular fractionation and RNA fluorescence in situ hybridization (FISH) assays, endogenous enrichment and abundant signals of *circ-CUX1* were observed in the nucleus of IMR32 cells, which was further confirmed by ectopic expression of *circ-CUX1* (Fig 2E and F). In this study, we observed the circularization efficiency of *circ-CUX1* vector by Northern blot, which indicated high levels of exogenous *circ-CUX1* than linear unspliced transcript (Appendix Fig S5A). To rule out the potential effects of linear unspliced transcript, we established a *circ-CUX1* vector with mutant back-splicing elements (from AG-GT to AG-CC, referred as *circ-CUX1*-Mut), which generated high levels of linear unspliced form, but not circular form, of *circ-CUX1* (Appendix Fig S5A). However, transfection of *circ-CUX1*-Mut into IMR32, SH-SY5Y, LoVo, and PC-3 cells resulted in no significant alteration in *circ-CUX1* expression, *p200 CUX1* promoter activity, levels of *CUX1* isoforms *p200* and *p110*, and glycolytic process (Fig 2H and Fig EV2A-C). These results indicated that the observed effects were *circ-CUX1* dependent. In this revised manuscript, we have added the data in Fig 2E, F, H, Fig EV2A-C, Appendix Fig S5A-C, and described these clearly at pages 7-8.

6. The statement that *circ-CUX1* exerts its role at the transcriptional level should be more clearly demonstrated. The authors should demonstrate that the activity is not elicited at level of *CUX1* mRNA stability (analysis of *CUX1* mRNA levels upon actinomycin treatment); moreover, they should check whether *circ-CUX1* localizes, according to the model, at the sites of the *p200 CUX1* promoter or of its target genes' transcription (ChIRP assay or double RNA/DNA FISH).

Response: Good comments. In this study, we found that *circ-CUX1* facilitated the expression of *p200 CUX1* at transcriptional level. Thus, it is rational to observe the potential effects of *circ-CUX1* on stability of *CUX1* mRNA. To meet this end, we observed the changes of *p200 CUX1* transcript levels in IMR32 and SH-SY5Y cells with stable transfection of *circ-CUX1* or sh-*circ-CUX1*, and those treated with actinomycin D. Our findings revealed that ectopic expression or knockdown of *circ-CUX1* did not affect the stability of *CUX1* mRNA (Fig EV2D). We definitively agree with the reviewer that it is necessary to observe the association of *circ-CUX1* with *p200 CUX1* promoter or target gene transcripts. To answer this question, we performed the RNA pull-down and chromatin isolation by RNA purification (ChIRP; *Methods Mol Biol*, 2016, 1480: 115-123) assays using biotin-labeled oligonucleotide probes targeting junction site of *circ-CUX1*, which indicated that *circ-CUX1* was associated with EWSR1 and MAZ protein, and promoters of target genes (*CUX1*, *S100A9*, *MUC4*, *KLF10*, or *TXNIP*), but not with transcripts of downstream gene (*p200 CUX1*, *ENO1*, *GPI*, *PGK1*, *S100A9*, *MUC4*, *KLF10*, or *TXNIP*) in SH-SY5Y cells (Appendix Fig S8A). These results demonstrated that *circ-CUX1* exerted its action through regulating gene transcription. In this revised manuscript, we have added the data in Fig EV2D and Appendix Fig S8A, and described these clearly at pages 7-8 and 11.

7. Fig. 4A: in order to have reliable results about *circ-CUX1* associated proteins, a native or crosslinked (preferred in order to exclude spurious interactions in solution) pulldown of endogenous *circ-CUX1* should be performed. Moreover, the inverse experiment (RIP or CLIP) should confirm the interaction also in non-transfected cells. The pulldown should also be specific, or at least enriched for

the circular form of CUX1, rather than the linear CUX1 mRNA. Therefore, the levels of both the circRNA and the mRNA should be checked in each pull-down or IP.

Response: In this study, to explore the protein partner of *circ-CUX1*, we performed RNA pull-down and mass spectrometry assays using biotin-labeled probe generated by ligation of linear transcript *in vitro* (Nucleic Acids Res, 2015, 43: 2454-2465). We definitively agree with the reviewer that in addition to these methods, it is necessary to observe native pull-down of endogenous *circ-CUX1*. Thus, we applied oligonucleotide probes targeting junction site of *circ-CUX1* for these studies. Mass spectrometry revealed 47 proteins consistently pulled down by exogenous *circ-CUX1* and antisense probe targeting endogenous *circ-CUX1*, but not by linear transcript or sense probe (Fig 4A), and 18 of them were RNA-binding protein (RBP) defined by RBPDB (<http://rbpdb.ccb.utoronto.ca>). Further comprehensive analysis of protein interacting with transcription factors of *p200 CUX1* promoter revealed by Genomatix and BioGRID database (<https://thebiogrid.org>) indicated three potential *circ-CUX1*-interacting partners (Fig 4A), including EWSR1, ELAV like RNA binding protein 1 (ELAVL1), and synaptotagmin binding cytoplasmic RNA interacting protein (SYNCRIP). Further validating RNA pull-down assay revealed the physical interaction of *circ-CUX1* with EWSR1, but not with ELAVL1 or SYNCRIP, in non-transfected IMR32 cells (Fig 4B). To confirm the specificity of pull-down and immunoprecipitation assays, we also observed the enrichment of circular (*circ-CUX1*) or linear (*p200* and *CASP*) forms of *CUX1*. The results indicated that *circ-CUX1* probes specifically pulled down endogenous or exogenous *circ-CUX1*, but not *p200 CUX1* or *CASP* transcript (Fig 4B), while EWSR1 protein bound to *circ-CUX1*, rather than *p200 CUX1* and *CASP* transcript (Fig 4C and F). In this revised manuscript, we have added the data in Fig 4A-C and F, and described these clearly at page 9.

8. Fig. 4C: it is very difficult to conclude about the colocalization of circ-CUX1 and the protein. In case a 3D-confocal reconstitutions should be provided. The image is quite surprising in terms of number of circ-CUX1 molecules, it is not clear from the legends whether it refers to an OE experiment. If the image corresponds to endogenous circ-CUX1 levels, they are very high. Have the authors checked what is the abundance of the endogenous RNA? Considering the levels of circ-CUX1, and in order to prove the model, a ChIRP assay or double DNA/RNA FISH should be provided.

Response: We are sorry for this confusing description of results. The presented images indicated the localization of *circ-CUX1* and EWSR1 in IMR32 cells stably over-expressing *circ-CUX1*. We definitively agree with the reviewer that it is necessary to measure the endogenous co-localization of *circ-CUX1* and EWSR1 protein with 3D-confocal reconstitution. To meet this end, the top or side images and 3D reconstitution of dual RNA fluorescence in situ hybridization (RNA-FISH) and immunofluorescence assay were provided to show the same spatial distribution of *circ-CUX1* (red) and EWSR1 (green) in IMR32 cells and those stably transfected with *circ-CUX1* (Fig 4D and Movie EV1). The results indicated endogenous co-localization of *circ-CUX1* and EWSR1 in IMR32 cells, which was facilitated by transfection of *circ-CUX1* (Fig 4D and Movie EV1). In addition, RNA pull-down and chromatin isolation by RNA purification (ChIRP, Methods Mol Biol, 2016, 1480: 115-123) assays were performed using biotin-labeled circRNA junction probe. The results indicated that *circ-CUX1* antisense probe specifically recognized endogenous *circ-CUX1*, but not *p200 CUX1* or *CASP* transcript, in IMR32 cells (Fig 4B). *Circ-CUX1* was associated with EWSR1 and MAZ protein, and promoters of target genes (*CUX1*, *S100A9*, *MUC4*, *KLF10*, or *TXNIP*), but not with transcripts of downstream genes (Appendix Fig S8A). In this revised manuscript, we have added the data in Fig 4D, Movie EV1, and Appendix Fig S8A, and described these clearly at pages 9 and 11.

9. Fig. 6C-D: in order to appreciate the effect of the synthetic peptide on MAZ transactivation and S100A9, MUC4, KLF10 and TXNIP proteins, a western blot similar to the ones in Fig. 5H and S10C should be shown. Have the authors tested the specificity of the effect of the peptide on other RNP complexes?

Response: We definitively agree with the reviewer that western blot assay is necessary to observe the effects of synthetic peptides on target gene expression *in vitro*. To meet this end, protein levels of EWSR1, MAZ, CUX1, ENO1, GPI, PGK1, S100A9, MUC4, KLF10, and TXNIP were detected in NB cells treated with CTLP or EIP-22. The results indicated that in consistent with *in vivo* studies, administration of EIP-22 altered the expression of *circ-CUX1* downstream genes in IMR32 and SH-SY5Y cells (Appendix Fig S9D). We also agree with the reviewer's opinion that it is necessary to test the specificity of synthetic peptides by observing their effects on other RNA ribonucleoprotein (RNP)

complexes. The results from RNA pull-down assay and cross-linking RIP assays indicated that EIP-22 treatment abolished the interaction between *circ-CUX1* and EWSR1, but not that of *pri-miR-222* and EWSR1 (Nucleic Acids Res, 2017, 45: 12481-12495) or *circACCI* and AMP-activated protein kinase beta 1 (AMPK β 1; Cell Metab, 2019, 30: 157-173) (Fig 6C and D). In this revised manuscript, we have added the data in Appendix Fig S9D, Fig 6C and D, and described these clearly at page 12.

10. Introduction - page 3: "For example, circRNA antisense to cerebellar-degeneration-related protein 1 (CDR1as) harbors 63 conserved miRNA binding sites for miR-7". Nota that actually, in Piwecka et al., 2017 the model of CDR1as as a sponge has been revised. This point should be discussed appropriately.

Response: According to studies by Piwecka *et al.*, cerebellar-degeneration-related protein 1 (*Cdr1as*) harbors 70 binding sites for miR-7, and directly binds miR-7 to regulate its transport in neurons, while miR-671 interacts with and slices *Cdr1as* to release its miR-7 cargo, suggesting a sophisticated regulatory network between circRNAs and miRNAs (Science, 2017, 357: 6357). In this revised manuscript, we have updated the reference and described these clearly at page 3.

11. Cell lines: the authors should specify in the text and figure legends which are the "NB cell lines" used in the work, to better understand the results.

Response: In this revised manuscript, we have described the exact NB cell lines in text and figure legends as required.

12. S1B: "The transcript levels of p200 CUX1, but not of CASP, were higher than those in normal fetal adrenal medulla (P<0.05, Appendix Fig S1A), especially in NB cases with poor stroma (P<0.0001) or advanced INSS stages (P=0.0081), without association with MYCN amplification (P=0.3511, Appendix Fig S1B)". From the boxplot, it seems that CUX1 is higher in cases with MYCN amplification ("Yes" column). Same comment for S4F.

Response: In this study, with addition of 12 cases, we observed the transcript levels of *p200 CUX1* and *CASP* in tumor tissues of total 54 NB cases. Although higher levels of *CUX1* were observed in NB cases with *MYCN* amplification, than those without *MYCN* amplification, the difference was lack of statistical significance (Appendix Fig S1B). In addition, patients with high *circ-CUX1* expression had lower survival probability (Fig EV1F). Due to limited size of cohort, the prognostic value of *circ-CUX1* and *CUX1* and their association with *MYCN* amplification in NB warrant further investigation. In this revised manuscript, we have added the data into Appendix Fig S1B and Fig EV1F, and described these clearly at pages 5, 7 and 16.

13. Fig. 2B: gel showing circ-CUX1 amplification should not be cropped, in order to see concatemers generated by the amplification of a circRNA-derived product of reverse transcription.

Response: We definitively agree with the reviewer that it is necessary to show full images of *circ-CUX1* amplification. In this revised manuscript, we have provided the full gel images in Fig 2B and described these clearly at page 6.

14. Fig. 2D: is it sure that the RNaseR was used 3 U·mg⁻¹ and not 3U·ug⁻¹ of RNA?

Response: We are sorry for this spelling mistake. In this revised manuscript, we have corrected this description into 3 U· μ g⁻¹ at pages 17, 20 and 34.

15. Fig. 4: there is a problem in how the panels are labeled in the figure and in the text.

Response: We are sorry for this labeling mistake. In this revised manuscript, we have double checked the correctness of panel labeling in figure and text as required.

Referee #2:

1. There are many bar graphs throughout the entire data presented in this manuscript. However, it is very difficult to distinguish the differences among veracious experimental groups due to the current drawing. The authors must re-scale most if not all bar graphs, in particular the y-axis to allow readers to clearly see the differences. For example, in Figure 1D, 1E, 1F, and 1G the full scale of Y-axis should be 4.0 or 80 (1F) instead of current ones. The revised bar graphs will allow one to observe the effects of *shCUX1* comparing with controls. The labels can be moved or placed on other places instead of at the top portion of the bar graphs. Similar revisions should also be made in bar graphs in other figures.

Response: We appreciate the reviewer's positive comments and revision guidance on our manuscript. We definitively agree with the reviewer that it is necessary to re-scale most bar graphs throughout the manuscript. In this revised manuscript, we have changed most scale of bars to make them clearly to show the differences, and placed the labels above the bar graphs in Fig 1B, D, E and G, Fig 3E, Fig 4C, Fig 5E-G, Fig 6D, Fig EV2A-C and E, Appendix Fig S1C, Fig S2A and D, Fig S5, Fig S6A, Fig S8A-D, Fig S9E-G, Fig S10A and C, and Fig S11D.

2. On page 11, the description of data in Figure 6C is an overstatement. Data in Figure 6C showed that treatment of EIP-22 peptide did not "abolish" but reduced the interaction of *circ-CUX1* with *EWSR*.

Response: We are sorry for this improper use of word. In this revised manuscript, we have corrected "abolished" into "reduced" at page 12 as required.

3. In Figure S2B, the authors should also analyze the relationships of genes expression of these four genes with patient survival using TCGA or another well annotated and curated public dataset. Similar concerns also go to data in Figure S4E and S4F. The clinical cases of NB tumors are too low to draw the conclusion. For example, in S4E, the last graph, one box only has a total of 9 NB samples. This number is too small to draw statistical conclusion.

Response: Good comments. We definitively agree with the reviewer that more public datasets should be analyzed to better understand the relationship of gene expression with survival of tumor patients. To meet this end, we have analyzed public datasets derived from Gene Expression Omnibus (GEO), The Cancer Genome Atlas (TCGA), and Kaplan-Meier plotter (<http://kmplot.com/>) databases as required. The results indicated that expression levels of *CUX1*, *ENO1*, *GPI*, and *PGK1* were associated with poor survival of tumor with neuroblastoma, colon cancer, prostate cancer, gastric cancer, or breast cancer (Appendix Fig S3). However, the survival data of *circ-CUX1* were not available in public datasets. In this study, with addition of 12 cases, we observed the *circ-CUX1* expression in total 54 NB cases. Although there were significant findings, the statistical conclusion warrants further investigated by a larger cohort of NB cases. In this revised manuscript, we have added data in Fig 2G, Appendix Fig S1B, Fig S3, Fig EV1E and F, Fig S7A and B, described these clearly at pages 5-7 and 10-11, and discussed the limitation at page 16.

Referee #3:

1. Does the combination of glycolysis inhibitors and anti *circ-CUX1* targeting synergize against NB tumor growth?

Response: We appreciate the reviewer's positive comments and revision guidance on our manuscript. Previous studies show that 2-deoxyglucose (2-DG), a glucose analog competes with glucose uptake, is able to inhibit activity of hexokinase II (HK2) and GPI, decrease production of ATP and lactate, and exert anti-tumor effects (Cancer Lett, 2014, 355: 176-183). As another known glycolysis inhibitor, 3-bromopyruvate (3-BP) is a promising anti-cancer compound that inhibits the activity of HK2 and glyceraldehyde-3-phosphate dehydrogenase (GAPDH), thereby reducing ATP production of cancer cells (J Bioenerg Biomembr, 2012, 44: 17-29). Since the process of aerobic glycolysis is catalyzed by many enzymes, and combining with our evidence that *CUX1*-generated circular RNA (*circ-CUX1*) facilitated the expression of glycolytic genes, enolase 1 (*ENO1*), glucose-6-phosphate isomerase (*GPI*), and phosphoglycerate kinase 1 (*PGK1*), we definitively agree with the reviewer that it is necessary to

investigate the potential synergistic effects of glycolysis inhibitors and anti-circ-CUX1 peptide (EIP-22) on the growth and aggressiveness of NB cells. In MTT colorimetric, soft agar, and matrigel invasion assays, EIP-22 treatment synergized the suppressing effects of glycolysis inhibitors, 2-DG and 3-BP, on the viability, growth, and invasion of IMR32 and SH-SY5Y cells (Appendix Fig S9E-G). In this revised manuscript, we have added data in Appendix Fig S9E-G, and described these clearly at pages 12 and 16.

2. Are the changes in ATP levels observed deriving only from glycolysis? Would be interesting to demonstrate that there are not disfunctions.

Response: Good comments. In this study, we found that ectopic expression of *p110 CUX1* increased the glucose uptake, lactate production, and ATP levels of NB cells (Appendix Fig S2A). Meanwhile, *circ-CUX1* facilitated the *CUX1* expression at transcriptional level. We definitively agree with the reviewer that it is necessary to investigate whether *CUX1*- or *circ-CUX1*-induced changes of ATP levels are derived from glycolysis. To meet this end, we applied the glycolysis inhibitor 2-DG in these studies, and found that 2-DG treatment abolished the increase of glucose uptake, lactate production, ATP levels of IMR32 cells induced by *p110 CUX1* or *circ-CUX1* over-expression (Appendix Fig S2A and Fig S6A), suggesting that glycolysis was the main source of ATP synthesis induced by *CUX1* and *circ-CUX1*. In this revised manuscript, we have added data in Appendix Fig S2A and Fig S6A, and described these clearly at pages 6 and 8.

2nd Editorial Decision

14 October 2019

Thank you for the submission of your revised manuscript to EMBO Molecular Medicine. We have now received the enclosed report from the two of the three referees who were asked to re-assess it. Since their recommendations are quite similar, I prefer to make a decision now rather than further delaying the process. As you will see the reviewers are now overall supportive and I am pleased to inform you that we will be able to accept your manuscript pending minor editorial amendments.

Please submit your revised manuscript within two weeks. I look forward to seeing a revised form of your manuscript soon.

***** Reviewer's comments *****

Referee #2 (Remarks for Author):

In this revised manuscript, the authors have satisfactorily addressed all the comments by me and other two reviewers with new data and corresponding changes throughout the manuscript. This revised study is sufficient for its publication in EMM.

Referee #3 (Comments on Novelty/Model System for Author):

The experimental strategies used by the authors are adequate and the results are convincing and original.

Referee #3 (Remarks for Author):

The authors have provided convincing evidences on the previously raised comments. At this stage the paper is suitable for publication without further revision.

2nd Revision - authors' response

15 October 2019

Authors made the requested editorial changes.

Corresponding Author Name: Qiangsong Tong, Liduan Zheng

Manuscript Number: EMM-2019-10835